# Large Language Model Is Not a Good Few-shot Information *Extractor*, but a Good *Reranker* for Hard Samples!

**Yubo Ma[1], Yixin Cao[2], YongChing Hong[1], Aixin Sun[1]**

[1] S-Lab, Nanyang Technological University
[2] Singapore Management University
yubo001@e.ntu.edu.sg

## Abstract

Large Language Models (LLMs) have made remarkable strides in various tasks. Whether LLMs are competitive few-shot solvers for information extraction (IE) tasks, however, remains an open problem. In this work, we aim to provide a thorough answer to this question. Through extensive experiments on nine datasets across four IE tasks, we demonstrate that current advanced LLMs consistently exhibit inferior performance, higher latency, and increased budget requirements compared to fine-tuned SLMs under most settings. Therefore, we conclude that LLMs are not effective few-shot information extractors in general [1]. Nonetheless, we illustrate that with appropriate prompting strategies, LLMs can effectively complement SLMs and tackle challenging samples that SLMs struggle with. And moreover, we propose an adaptive *filter-then-rerank* paradigm to combine the strengths of LLMs and SLMs. In this paradigm, SLMs serve as filters and LLMs serve as rerankers. By prompting LLMs to rerank a small portion of difficult samples identified by SLMs, our preliminary system consistently achieves promising improvements (2.4% F1-gain on average) on various IE tasks, with an acceptable time and cost investment. Our code is available at https://github.com/mayubo2333/LLM-IE.

## 1 Introduction

Large Language Models (LLMs, Brown et al. 2020; Chowdhery et al. 2022; Touvron et al. 2023) have shown remarkable abilities on various NLP applications such as factual question answering (Yu et al., 2023; Sun et al., 2023), arithmetic reasoning (Chen et al., 2022a; Qian et al., 2023) and logical reasoning (Jung et al., 2022; Pan et al., 2023). Given the reasoning, memorization, instruction-following and few-shot adaption capabilities emerging from

LLMs, it prompts a compelling question: Can LLMs be used to boost performance in few-shot information extraction (IE) tasks?

To answer this question, we conduct an extensive empirical study to compare the performance between LLMs using *in-context learning* [2] (ICL) and *fine-tuned* Small Language Models (SLMs). We fairly evaluate SLMs-based and LLMs-based methods across nine datasets spanning four common IE tasks: (1) Named Entity Recognition, (2) Relation Extraction, (3) Event Detection and (4) Event Argument Extraction. For each dataset, we explored four to six settings to encompass typical low-resource extents, from 1-shot to 20-shot or even more. Given the potential sensitivity of LLMs' performance to the prompt context, we meticulously considered variations in instruction, demonstration number and selection strategy, prompt format, *etc*. Our study reveals that LLMs excel over SLMs only when annotations are extremely limited, *i.e.,* both label types[3] and the samples[4] per label are extremely scarce. With more (*e.g.,* hundreds of) samples, SLMs significantly outperform LLMs. Furthermore, LLMs incur greater inference latency and costs than fine-tuned SLMs. Hence, we claim that **current LLMs are not good few-shot information extractors in general**.

We further investigate whether LLMs and SLMs exhibit different abilities to handle various types of samples. We categorize samples according to their difficulty measured by SLMs' confidence scores, and compare LLMs' and SLMs' results within each group. We find that **LLMs are good at hard samples, though bad at easy samples**. We posit that the knowledge and reasoning abilities in LLMs enable them to handle hard samples (which are sim-

---

[1] A more precise assertion is that *current LLMs, with vanilla prompting setting and without IE-specific fine-tuning, are not good few-shot information extractors in general.*

[2] All LLMs discussed in this paper are not fine-tuned, and results for LLMs are based on in-context learning.

[3] Label types denote *entity/relation/event/role types* in different tasks. We use them interchangeably there-in-after.

[4] Samples refer to (i) demonstrations in ICL of LLMs, or (ii) training samples for SLMs' fine-tuning.

ply beyond SLMs' capabilities) well. Nevertheless, LLMs demonstrate strong predisposition to false-positive predictions on negative samples. Since most negative samples are easy samples (which could be solved readily by SLMs), the performance of LLMs on easy samples sometimes collapses and are usually much worse than fine-tuned SLMs.

Leveraging these findings, we pursue an approach to incorporate LLMs and SLMs within a single system and combine their merits. To this end, we propose a novel *filter-then-rerank* framework. The basic idea is that SLMs serve as a filter and LLMs as a reranker. Specifically, SLMs initially predict and determine the difficulty of each sample. If the sample is a hard one, we further pass the top-$N$ most-likely candidate labels from SLMs to LLMs for reranking. Otherwise we view the prediction from SLMs as the final decision. By providing easy/hard samples with different solution strategies, our system utilizes each model's strengths to complement each other. Also, it reranks only a small subset of samples and minimizes the extra latency and budgets for calling LLMs. With a modest cost increase, our framework yields a consistent F1 improvement, averaging 2.4% higher than previous methods on various few-shot IE tasks. To the best of our knowledge, this is the first successful attempt to use LLMs to enhance few-shot IE tasks.

## 2 Related Work

### 2.1 LLMs for Information Extraction

Recent studies have increasingly explored Information Extraction (IE) tasks using LLMs. Drawing inspiration from instruction tuning (Wei et al., 2022a), several methods (Wadhwa et al., 2023; Wang et al., 2023a; Lu et al., 2023) transform annotated samples into instruction-answer pairs and then fine-tune LLMs, such as FlanT5 (Chung et al., 2022), on them. Nonetheless, this method necessitates a vast range of samples with diverse schemas and often yields suboptimal results in low-resource scenarios. In the context of few-shot IE tasks, prevalent strategies bifurcate into two main streams. The first approach perceives LLMs as efficient annotators (Ding et al., 2023; Josifoski et al., 2023). In these methods, they produce a plethora of pseudo-labeled samples through LLMs and leverage the enhanced annotations to train SLMs. Conversely, the latter approach employs LLMs in inference using the ICL paradigm, which is the focus of our subsequent discussion.

### 2.2 Few-shot IE with ICL

Regarding few-shot IE tasks, recent studies intensively compare the performance between SLMs and LLMs but yield inconsistent conclusions. Some studies favor LLMs as competent few-shot extractors (Agrawal et al., 2022; Wang et al., 2023b; Li et al., 2023; Zhang et al., 2023a; Wadhwa et al., 2023), while others dispute this claim (Jimenez Gutierrez et al., 2022; Qin et al., 2023; Wei et al., 2023; Gao et al., 2023). This discrepancy leaves the question of *whether LLMs perform competitively on few-shot IE tasks* unresolved, thus hindering the advances of this domain.

We attribute such disagreement to the absence of an comprehensive and unified benchmark. Existing studies usually vary in tasks, datasets, and few-shot settings. Furthermore, some studies rely on overly simplistic datasets (Jimenez Gutierrez et al., 2022; Li et al., 2023) and may exaggerate the effectiveness of LLMs. Driven by these findings, our research undertakes comprehensive experiments across four IE tasks, nine datasets with various schema complexities (from coarse-grained to fine-grained) and low-resource settings.

In addition to the empirical study, we develop an innovative *filter-then-rerank* paradigm to combine the strengths of both LLMs and SLMs. It utilizes prompting strategies akin to QA4RE (Zhang et al., 2023a), transforming IE tasks into multi-choice questions. However, our method stands apart by integrating SLMs and LLMs within a single framework. This incorporation (1) enables our paradigm applicable to various IE tasks by providing candidate spans in the text and (2) achieves promising performance under low-resource IE scenarios.

## 3 Large LMs v.s. Small LMs

In this section, we compare the performance between LLMs and SLMs to evaluate whether LLMs perform competitively.

### 3.1 Task, Dataset and Evaluation

We run experiments on nine widely-used datasets across four IE tasks. (1) Named Entity Recognition (NER): CONLL03 (Tjong Kim Sang and De Meulder, 2003), OntoNotes (Weischedel et al., 2013) and FewNERD (Ding et al., 2021). (2) Relation Extraction (RE): TACRED (Zhang et al., 2017) and TACREV (Alt et al., 2020). (3) Event Detection (ED): ACE05 (Doddington et al., 2004), MAVEN (Wang et al., 2020) and ERE (Song et al.,

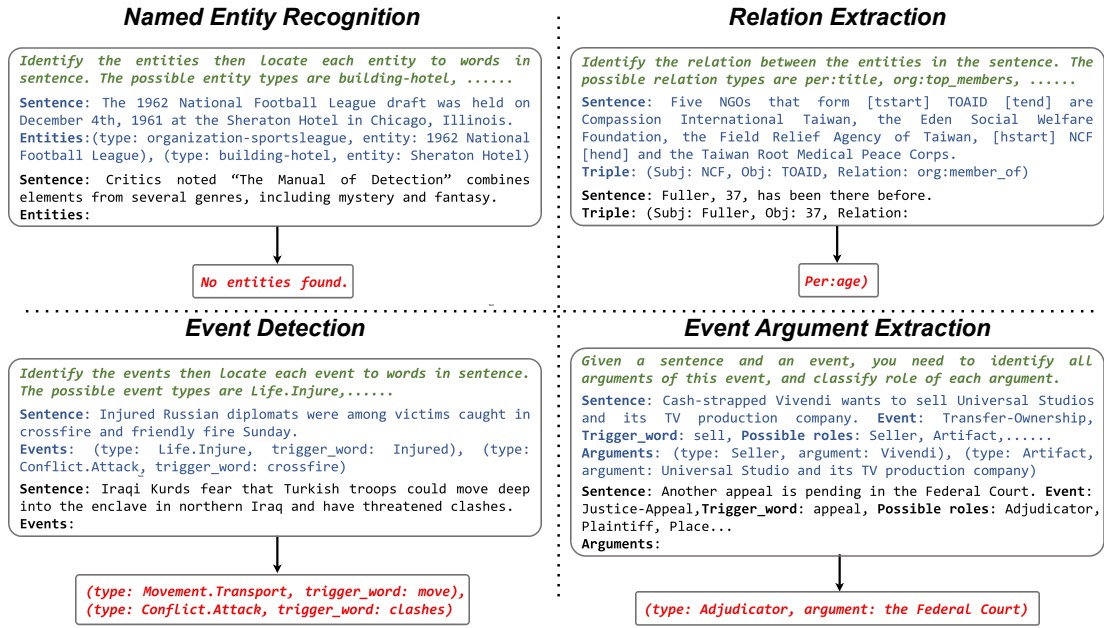

**Named Entity Recognition**

> *Identify the entities then locate each entity to words in sentence. The possible entity types are building-hotel, ......*
> **Sentence**: The 1962 National Football League draft was held on December 4th, 1961 at the Sheraton Hotel in Chicago, Illinois.
> **Entities**:(type: organization-sportsleague, entity: 1962 National Football League), (type: building-hotel, entity: Sheraton Hotel)
> **Sentence**: Critics noted "The Manual of Detection" combines elements from several genres, including mystery and fantasy.
> **Entities**:

*No entities found.*

**Relation Extraction**

> *Identify the relation between the entities in the sentence. The possible relation types are per:title, org:top_members, ......*
> **Sentence**: Five NGOs that form [tstart] TOAID [tend] are Compassion International Taiwan, the Eden Social Welfare Foundation, the Field Relief Agency of Taiwan, [hstart] NCF [hend] and the Taiwan Root Medical Peace Corps.
> **Triple**: (Subj: NCF, Obj: TOAID, Relation: org:member_of)
> **Sentence**: Fuller, 37, has been there before.
> **Triple**: (Subj: Fuller, Obj: 37, Relation:

*Per:age*

**Event Detection**

> *Identify the events then locate each event to words in sentence. The possible event types are Life.Injure,......*
> **Sentence**: Injured Russian diplomats were among victims caught in crossfire and friendly fire Sunday.
> **Events**: (type: Life.Injure, trigger_word: Injured), (type: Conflict.Attack, trigger_word: crossfire)
> **Sentence**: Iraqi Kurds fear that Turkish troops could move deep into the enclave in northern Iraq and have threatened clashes.
> **Events**:

*(type: Movement.Transport, trigger_word: move), (type: Conflict.Attack, trigger_word: clashes)*

**Event Argument Extraction**

> *Given a sentence and an event, you need to identify all arguments of this event, and classify role of each argument.*
> **Sentence**: Cash-strapped Vivendi wants to sell Universal Studios and its TV production company. **Event**: Transfer-Ownership, **Trigger_word**: sell, **Possible roles**: Seller, Artifact,......
> **Arguments**: (type: Seller, argument: Vivendi), (type: Artifact, argument: Universal Studio and its TV production company)
> **Sentence**: Another appeal is pending in the Federal Court. **Event**: Justice-Appeal,**Trigger_word**: appeal, **Possible roles**: Adjudicator, Plaintiff, Place...
> **Arguments**:

*(type: Adjudicator, argument: the Federal Court)*

Figure 1: Examples of prompts used. The green, blue and black parts in the top boxes represent the instruction, demonstration (demo) and test sentence in the prompt respectively. The red parts represent the outputs from LLMs. We plot only 1 example for convenience of visualization. The actual demo number is usually much larger than 1.

2015). (4) Event Argument Extraction (EAE): ACE05, ERE and RAMS (Ebner et al., 2020). With label numbers ranging from 4 to 168, we assess LLMs' performance under different schema complexities. See their details in Appendix A.1.

**Few-shot Set** We construct few-shot datasets from the original datasets above. For training and validation set, we adopt $K$-shot sampling strategy, *i.e.,* sampling $K$ samples for each label type. See more details in Appendix A.2. For test set, we downsample their original test sets to reduce the cost of LLMs. We randomly sample 500 sentences for RE tasks, and 250 sentences for other task. We ensure that each label has at least one corresponding sample to avoid the absence of rare labels.

**Evaluation** We adopt micro-F1 score in NER, RE and ED tasks. For EAE task, we follow previous work (Wang et al., 2023b) and adopt head-F1 score, which merely considers matching of the head word rather than the whole content of a text span. We report averaged score w.r.t 5 sampled train/validation sets unless otherwise stated.

### 3.2 Small Language Models

We adopt five supervised methods to evaluate the abilities of SLMs. (1) Vanilla fine-tuning for all tasks, (2) FSLS (Ma et al., 2022a) for NER and ED tasks, (3) KnowPrompt (Chen et al., 2022b) for RE task, (4) PAIE (Ma et al., 2022b) for EAE task, and

(5) UIE (Lu et al., 2022c) for all tasks. See their details in Appendix B.

### 3.3 Large Language Models

Detailed in Appendix C, we evaluate the ICL abilities of LLMs. Given labeled sentences $D = \{(s_i, y_i)\}$ and a test sentence $s$, our goal is to predict structured information $y$ from $s$ using a frozen LLM $\mathcal{L}$. We feed LLM with prompt $\mathcal{P}_{\mathcal{E},I,f}(D, s)$:

$$\mathcal{P}_{\mathcal{E},I,f}(D, s) = [I; f(\mathcal{E}(D, s)); f(s)] \quad (1)$$

We give examples of prompts on four IE tasks in Figure 1. The prompts consist of three parts: instruction $I$ (color in green in Figure 1), demonstration $f(\mathcal{E}(D, s))$ (demo; color in blue) and the question $f(x)$ (color in black). Here $\mathcal{E}$ denotes demo selector and $\mathcal{E}(D, s) \subset D$ denotes selected sentences as the demo to predict $s$. Prompt format $f$ [5] refers to the template which converts demo $\mathcal{E}(D, s)$ and sample $s$ to input context for LLMs. Then LLM generates $f(y)$ (color in red) from which we could readily parse the extraction results $y$.

**Models** $\mathcal{L}$: We explore six LLMs from two sources. (1) OpenAI models [6]: we employ Chat-

---

[5] We slightly abuse the notation $f$ to allow $s$, $y$ and $\{(s, y)\}$ as the input for simplicity.

[6] The versions of model we use are: gpt-3.5-turbo-0301,

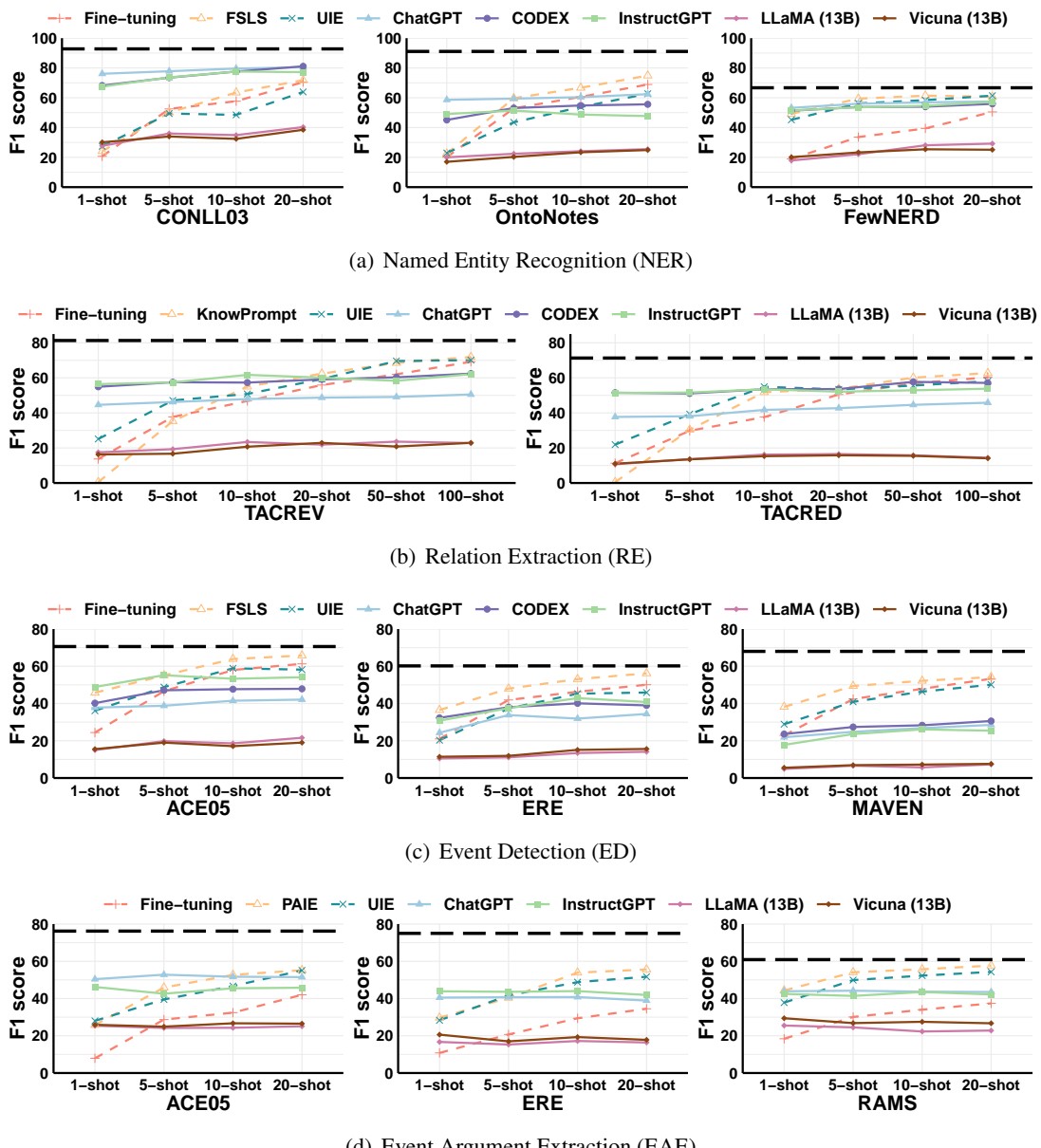

Figure 2: Overall results of SLM-based methods (dashed lines) and LLM-based methods (solid lines) on nine datasets across four IE tasks. The black, horizontal dashed lines represent the SoTA performance on full dataset.

GPT, CODEX (Chen et al., 2022a) and Instruct-GPT (Ouyang et al., 2022) for main experiments. We also evaluate GPT-4 in Appendix D.3. (2) Open-source models: we use LLaMA-13B (Tou-vron et al., 2023) and its instruction-tuned counterpart, Vicuna-13B (Chiang et al., 2023).

**Instruction** $I$: The instruction (1) describes the task and (2) enumerates all possible labels for reference. we adopt instructions shown in Figure 1.

**Demo selector** $\mathcal{E}$: The maximum input length of

LLMs usually limits the sentence number in demos even under few-shot settings. Therefore for each test sentence $s$, we demand a demo retriever $\mathcal{E}(D, s)$ which selects a small subset from $D$ as the sentences in demo. Following previous methods (Liu et al., 2022; Su et al., 2022), we retrieve demos according to their sentence embedding similarity to the test samples.

**Prompt format** $f$: We use simple textual templates to format the demos and the test sample in main experiments. For example, the template for NER is "Sentence: [S], Entities: ([type1], [entity1]), ([type2], [entity2])...".

---

code-davinci-002, text-davinci-003 and gpt-4-0314. Due to budget constraints, we execute InstructGPT and GPT-4 only once per setting. We do not conduct EAE task on CODEX since it had been unavailable at that time.

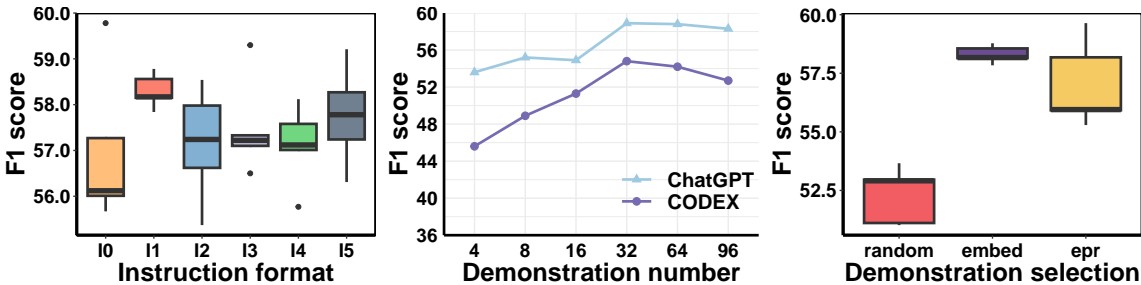

Figure 3: LLMs' performance w.r.t prompt variants on 20-shot FewNERD dataset. See full results on other datasets in Appendix E.2- E.5. **Left**: ChatGPT's performance (F1 Score) across six instruction variants. **Middle**: F1 Score changes over varying numbers of demo. **Right**: ChatGPT's performance across three demo selection strategies. Random: Random sampling. Embed: Sentence embedding. EPR: Efficient Prompt Retriever (Rubin et al., 2022).

## 3.4 Main Results

We summarize the main experimental outcomes in Figure 2, indicating that LLMs only outperform SLMs in environments with restricted labels and samples. Conversely, SLMs are generally more effective. Given (1) the practicality of fine-grained IE tasks and the manageable effort of obtaining 10-20 annotations per label and (2) the excessive time and budget demands of LLM inference, we conclude that LLMs are not as effective as supervised SLMs for few-shot IE tasks under real scenarios. We detail our findings as below.

**Performance w.r.t sample number.** The performance dynamics of SLMs and LLMs are influenced by variations in sample size. Under extremely low-resource (1-shot or 5-shot) settings, LLMs sometimes present superior performance than SLMs. Yet, LLMs tend to reach a performance plateau with only modest increases in sample size. Conversely, SLMs demonstrate marked performance enhancement as sample sizes grow. This trend is evident in Figure 2, where the SLM trajectories (represented by dashed lines) ascend more steeply compared to the LLM ones (solid lines).

**Performance w.r.t label number.** Compared with SLMs, LLMs tend to struggle on fine-grained datasets. For instance, LLMs perform *relatively* worse on MAVEN and RAMS datasets (with 168/139 labels) than on CONLL (4 labels only). Detailed quantitative results are shown in Appendix E.1, illustrating a clear negative correlation between the label number and the result disparity between LLMs and SLMs across various IE tasks.

**Comparisons among LLMs.** We observe performance variability among LLMs. (1) Open-source models, LLaMA and Vicuna, significantly lag behind proprietary LLMs across all few-shot IE tasks.

(2) Among proprietary LLMs, ChatGPT performs better on NER and EAE tasks, but poorer so on RE and ED tasks. InstructGPT and CODEX demonstrate comparable performance across these tasks. **LLMs show limited inference speed.** We compare the inference speed of different methods and show their results in Table 1. We observe that LLMs is much slower than SLMs since they have much more parameters, longer input contexts and extra response decay (if external APIs applied).

## 3.5 Analysis on Prompt Sensitivity

Previous work (Lu et al., 2022b) indicates that the efficacy of LLMs on specific tasks can be significantly influenced by the construction of the prompt. To ensure that LLMs' suboptimal outcomes are not erroneously ascribed to inappropriate prompt designs, we meticulously examine the impact of diverse prompt variations from four aspects, *i.e.,* instruction format, demo number, demo selector and prompt format. We leave comprehensive details of the variants and their results to Appendix E.2-E.5, and illustrate salient findings in Figure 3. Our findings include that (1) diverse instruction strategies yield comparable results in IE task; (2) increasing the number of samples in demonstrations does not unequivocally enhance performance; and (3) The selection strategy of demonstration matters, and retrieval based on sentence embedding

Table 1: The inference seconds over 500 sentences (run on single V100 GPU). Here LLaMA is extremely slow since we set batch size as 1 due to memory limit.

| Dataset (Task) | Roberta | T5 | LLaMA | CODEX |
|---|---|---|---|---|
| FewNERD (NER) | 2.8 | 39.4 | 1135.4 | 179.4 |
| TACREV (RE) | 1.4 | 45.6 | 1144.9 | 151.6 |
| ACE05 (ED) | 6.6 | 62.5 | 733.4 | 171.7 |

(what we used) proves sufficiently effective. Consequently, we believe that there unlikely exists a *lottery* prompt that substantially alters our conclusions that LLMs are not good few-shot IE solver.

### 3.6 Discussion: Why LLMs Fail to Obtain Satisfactory Performance on IE Tasks?

**Underutilized Annotations.** We notice that LLMs appear to benefit less from additional annotations, *i.e.,* more training samples and label types, than SLMs. We speculate that LLMs are constrained by ICL in two ways. (1) More samples: The number of effective samples for LLMs, those in demos, is limited by maximum input length. Moreover, we also observe LLMs' performance plateaus in some tasks before reaching this limit (see Appendix E.3). Meanwhile, SLMs can continually learn from more samples through supervised learning, widening the performance gap as annotated samples increase. (2) More labels: LLMs struggle with fine-grained datasets. It suggests a difficulty in understanding numerous labels and their subtle interactions merely from the given instruction and exemplars for LLMs. Also, the examples per label in demos decrease as label types increase.

**Unexplored Task format.** As stated in Zhang et al. (2023a), IE-related tasks are scarce in the widely-used instruction tuning datasets like Wei et al. (2022a) and Wang et al. (2022). Furthermore, the highly-flexible format of NER and ED tasks impair the ICL abilities [7]. Therefore it is likely that instruction-tuned LLMs are not well-acquainted with such IE-related task formats.

## 4 LLMs are Good Few-shot Reranker

### 4.1 Filter-then-rerank Paradigm

```
Read following sentences and identify what is the entity type
of "The New Yorker" quoted by <t>.
Sentence:
In 2004 Gourevitch was assigned to cover the 2004 U.S.
presidential election for "<t> The New Yorker <t>".
Candidate Choices:
(a)The New Yorker does not belong to any known entities.
(b)The New Yorker is a broadcast program.
(c)The New Yorker is a kind of written art.
(d)The New Yorker is a media/newspaper organization.
Analysis:
The New Yorker is a well-known American magazine that has
been published since 1925, and is primarily known for its
long-form journalism, commentary, and satire. It has a
reputation for publishing high-quality writing on a wide
variety of topics, including politics, culture, and the arts.
So The New Yorker is a media/newspaper organization.
Correct Answer: (d)
```

Figure 4: Multi-choice question (MCQ) prompt.

---

[7]These two tasks require unfixed numbers of (label, span) tuple. Furthermore, the length of each span is also unfixed.

To mitigate LLMs' drawbacks mentioned above, we propose a *filter-then-rerank* paradigm to integrate both SLMs and LLMs within the same system. This paradigm uses SLMs as filters to select the top-$N$ candidate labels, then LLMs rerank them to make final decisions. By using SLM-generated candidate answers, the focus of LLMs shifts from **sentence-level** (*i.e.,* identifying all entities/events in the sentence) to **sample-level** (*i.e.,* determining single entity/event candidate provided). Each question now corresponds to a single sample, allowing us to reframe prompts as multi-choice questions (MCQ; shown in Figure 4) problem. Under such format, each candidate label is converted to a choice by pre-defined templates. We claim *filter-then-rerank* paradigm is more likely to elicit the powers of LLMs and smoothly solve few-shot IE tasks because: (1) LLMs are more familiar with MCQ prompts than IE-format prompts (Zhang et al., 2023a). (2) This paradigm reduces the label scopes significantly, since $N$ is usually much smaller than fine-grained label numbers.

### 4.2 LLMs are *Hard* Sample Solver

Our *filter-then-rerank* paradigm, unfortunately, presents unsatisfactory performance (and even suffers longer latency since LLMs rerank candidates per sample). Given LLMs' abilities in memorization and reasoning, however, we still believe that LLMs are potential to solve **some**, if not most, IE samples effectively. We hypothesize that LLMs are more proficient than SLMs on *hard* samples. These samples are characterized by their requisite for external knowledge acquisition or sophisticated reasoning strategies, areas where LLMs can leverage their extensive parametric knowledge bases and inherent reasoning mechanisms. In contrast, SLMs often falter with such samples, constrained by their restricted modeling capacities.

We leverage an unsupervised metric from SLMs to evaluate the *difficulty* of samples. Given a sample $x$ in the sentence $s$, we define the highest probability across all labels as the confidence score:

$$\text{conf}(x) = \max_{l \in L} P_{SLM}(l|x; s) \qquad (2)$$

where $L$ denotes the label set and $P_{SLM}(l|x; s)$ the probability of a span $x$ (in the sentence $s$) referring to label $l$ computed by SLMs. We classify samples with low confidence scores as *hard* samples. Otherwise we view them as easy samples.

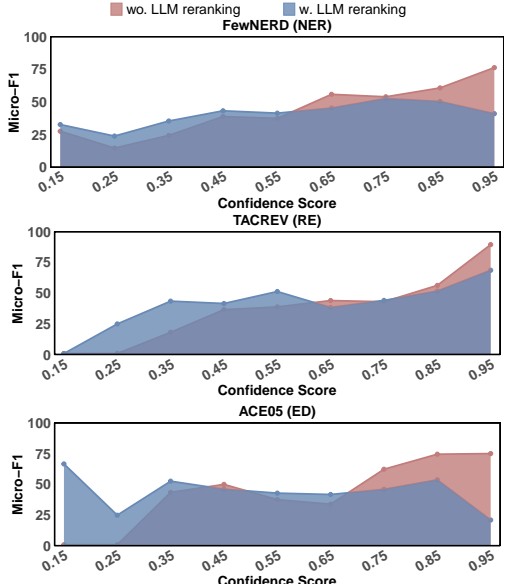

Figure 5: Relationship between confidence scores and performance with/without LLM reranking. We adopt `RoBERTa-large` as filter and InstructGPT as reranker.

We conduct experiments to confirm our hypothesis that LLMs excel on *hard* samples. We group samples by confidence scores and compare two methods within each group: (a) SLM-based methods without LLM reranking, and (b) SLMs as the filter and LLMs as the reranker. Method (b) differs from (a) by adding a single LLM to rerank the top-$N$ SLM predictions, using MCQ prompts.

The results in Figure 5 substantiate our assumption. (1) LLM-based reranking (blue lines) enhances performance on hard samples (left areas in the figure). We provide a detailed analysis of specific challenging instances where LLM rerankers prove advantageous in Appendix F.1. These instances demonstrate the efficacy of LLMs in harnessing external knowledge and complex reasoning to rectify erroneous predictions initially made by SLMs (red lines). (2) Conversely, LLM-based reranking impedes performance on easy samples (right areas), resulting in a significant degradation, particularly for very easy samples (rightmost areas). In conclusion, LLMs exhibit greater proficiency in handling hard samples compared to SLMs, yet they underperform relative to SLMs on easy samples.

### 4.3 Why LLMs Fail on Easy Samples

We investigate why LLMs (relatively) fail on easy samples in this section. As shown in Table 2, we observe significant higher negative sample ratios for easy samples across diverse IE tasks. In other

Table 2: Comparative ratios of negative to positive samples across various datasets and subsets. We set fixed threshold $\tau$ here for simplicity.

|  | FewNERD | TACREV | ACE05 |
|---|---|---|---|
| Overall | 5.88 | 3.03 | 38.2 |
| Easy samples ($\tau > 0.9$) | 9.44 | 3.21 | 44.0 |
| Hard samples ($\tau < 0.6$) | 1.28 | 2.68 | 1.36 |

words, most negative samples are easy samples for SLMs. Here we refer negative samples to those labeled as None. We speculate that the proficiency of SLMs with negative samples stems from their ability to adeptly discern apparent patterns during the fine-tuning stages. Therefore, SLMs could predict negative samples with (relatively) high confidence and accuracy. Due to LLMs' predisposition to false-positive predictions on negative samples, however, the performance of LLMs on easy samples collapses. We attribute such false-positive predictions to (1) hallucination and (2) span boundary mismatch. We detail such two kinds of mistakes with cases in Appendix F.2.

## 5 Adaptive Filter-then-rerank Paradigm

Above findings can be summarized as: (1) SLMs generally outperform LLMs, especially with more training samples and fine-grained labels. (2) SLMs are much more time- and cost-efficient. (3) LLMs serve as powerful rerankers on *hard* samples that challenge SLMs. Based on them, we propose a simple, efficient, and effective adaptive reranker that combines the strengths of SLMs and LLMs.

### 5.1 Method

Our *adaptive filter-then-rerank* approach, shown in Figure 6, uses supervised SLMs as a filter to make preliminary decisions. Samples with confidence scores exceeding threshold are viewed as easy samples otherwise hard ones. For easy samples, we retain SLM predictions as final results. For hard samples, top-$N$ predictions from SLMs are reranked via LLMs using ICL. Here LLMs employ MCQ prompts (Figure 4), containing demos and a sample to be reranked. The LLMs then generate the final answer and optionally provide an explanation.

### 5.2 Experimental Setup

We conduct experiments on FewNERD for NER task, TACREV for RE task and ACE05 for ED task. We employ top-performing SLM-based methods from Section 3 (FSLS or KnowPrompt) as the

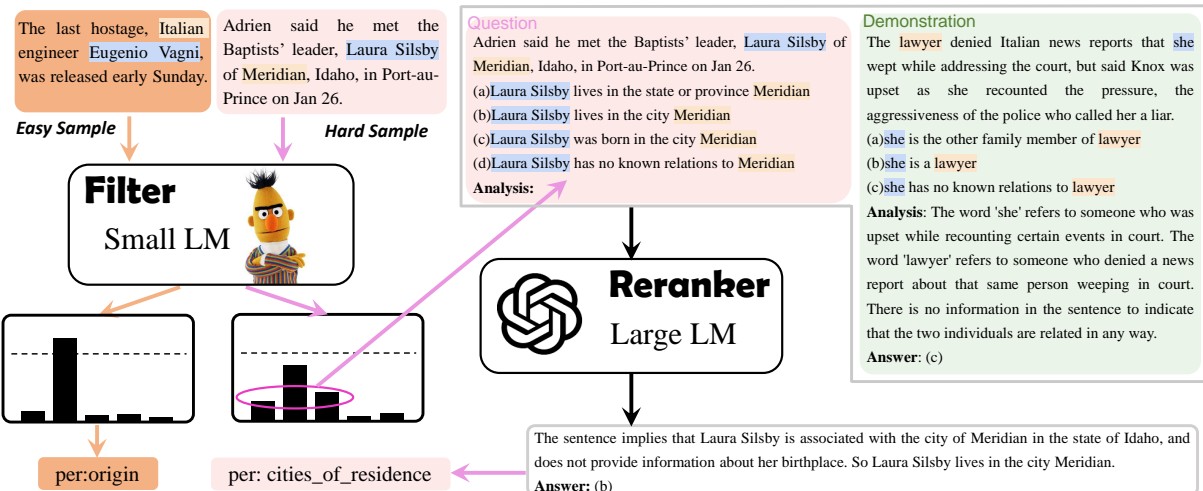

Figure 6: The overall architecture of our adaptive *filter-then-rerank* paradigm. We color easy samples in orange and hard samples in pink. For easy samples, the final predictions are exactly from the SLM-based methods. For hard samples, the top-$N$ predictions from SLMs are fed into LLMs as the format of multiple-choice questions (pink box). The question is paired with demos (green box). LLMs rerank these $N$ candidates and generate the final prediction.

filter, and Vicuna-13B, InstructGPT or GPT-4 as the reranker. The threshold $\tau$ to determine sample difficulty is optimized on the valid set. For hard sample, the top-3 SLM predictions and None (if not included) are feed to LLMs for reranking. Each LLM prompt has 4-shot demos. See demo examples in Appendix G.1. We follow templates in Lu et al. (2022a) for TACREV and carefully design others. See these templates in Appendix G.2. We adopt chain-of-thought reasoning (Wei et al., 2022b), *i.e.,* prefacing the answer with an explanation, to facilitate LLMs' reranking procedure.

**Baseline** We compare our method with two kinds of baselines to validate its effectiveness.
(1) LLMs with ICL: We follow the prompts in Section 3.3 and conduct experiments on three LLMs.
(2) Supervised SLMs: We follow previous SoTA methods shown in Section 3.4 (FSLS or Know-Prompt). We additionally combine two SLMs with ensemble or reranking approach (*i.e.,* replace the LLM with another SLM as the reranker) to verify that improvements from our SLM-LLM integrated system are not solely due to the ensemble effects.

### 5.3 Main Results

Table 3 shows that our *filter-then-rerank* method consistently improves performance across three datasets and nine settings. For instance, with InstructGPT, reranking provides an average F1 gain of 2.4% without SLM ensemble (Lines 4 vs. 7). Based on ensemble SLMs as the filter, our method still achieves 2.1% (Lines 5 vs. 8) gains on av-

erage. This confirms (1) the effectiveness of the LLM reranking and (2) its gains are different and (almost) orthogonal to the SLM ensemble.

### 5.4 Analysis

**Few makes big difference** Our method selectively reranks hard samples. Table 4 shows that (1) only a minor fraction (0.5%~10%) of samples are deemed hard and are reranked by LLMs. (2) Despite their limited quantity, reranking results in a substantial performance boost on these samples (10%~25% absolute F1 gains). This uplift on a small subset significantly enhances the overall performance.

**GPT-4 is more aggressive** From Tables 3 and 4, GPT-4 generally improves more on hard samples, yet InstructGPT surpasses GPT-4 in NER and RE tasks when evaluated overall. This discrepancy arises from GPT-4's aggressive reranking which introduces more true positives. InstructGPT, however, focuses more on reducing false positives.

**Few makes small cost** Figure 7 demonstrates that our method impressively reduces budget and latency by approximately 80%~90% compared to direct ICL. This reduction is due to (1) fewer LLM callings (only for hard samples) and (2) shorter prompts (fewer candidate labels and demos).

### 5.5 Ablation Study

We investigate the effectiveness of the modules in adaptive *filter-then-rerank* system by removing each of them in turn: (1) **CoT**: We exclude the explantion for each examples in demo. (2) **Demo**:

Table 3: Overall results of LLM-based ICL methods, SLM-based supervised methods, and our proposed *filter-then-rerank* (SLM+LLM) methods. The best results are in bold face and the second best are underlined. All results except InstructGPT and GPT-4 are averaged over 5 runs, and sample standard deviations are in the round bracket.

| | | FewNERD (NER) | | | TACREV (RE) | | | ACE (ED) | | |
|---|---|---|---|---|---|---|---|---|---|---|
| | | 5-shot | 10-shot | 20-shot | 20-shot | 50-shot | 100-shot | 5-shot | 10-shot | 20-shot |
| **LLM** | CODEX | 53.8(0.5) | 54.0(1.4) | 55.9(0.5) | 59.1(1.4) | 60.3(2.4) | 62.4(2.6) | 47.1(1.2) | 47.7(2.8) | 47.9(0.5) |
| | InstructGPT | 53.6(−) | 54.6(−) | 57.2(−) | 60.1(−) | 58.3(−) | 62.7(−) | 52.9(−) | 52.1(−) | 49.3(−) |
| | GPT-4 | - | - | 57.8(−) | - | - | 59.3(−) | - | - | 52.1(−) |
| **SLM** | Previous SoTA | 59.4(1.5) | 61.4(0.8) | 61.9(1.2) | 62.4(3.8) | 68.5(1.6) | 72.6(1.5) | 55.1(4.6) | 63.9(0.8) | 65.8(2.0) |
| | + Ensemble (S) | 59.6(1.7) | 61.8(1.2) | 62.6(1.0) | 64.9(1.5) | 71.9(2.2) | 74.1(1.7) | 56.9(4.7) | 64.2(2.1) | 66.5(1.7) |
| | + Rerank (S) | 59.4(1.5) | 61.0(1.7) | 61.5(1.7) | 64.2(2.3) | 70.8(2.3) | 74.3(2.2) | 56.1(0.3) | 64.0(1.0) | 66.7(1.7) |
| **SLM + LLM** | *Vicuna-13B* | | | | | | | | | |
| | + Rerank (L) | 60.0(1.8) | 61.9(2.1) | 62.2(1.4) | 65.2(1.4) | 70.8(1.6) | 73.8(1.7) | 56.9(4.0) | 63.5(2.7) | 66.0(2.6) |
| | + Ensemble (S) + Rerank (L) | 59.9(0.7) | 62.1(0.7) | 62.8(1.1) | 66.5(0.5) | 73.6(1.4) | 75.0(1.5) | 57.9(5.2) | 64.4(1.2) | 66.2(2.4) |
| | *InstructGPT* | | | | | | | | | |
| | + Rerank (L) | 60.6(2.1) | 62.7(0.8) | 63.3(0.6) | 66.8(2.6) | 72.3(1.4) | 75.4(1.5) | 57.8(4.6) | 65.3(1.7) | 67.3(2.2) |
| | + Ensemble (S) + Rerank (L) | **61.3**(1.9) | **63.2**(0.9) | **63.7**(1.8) | **68.9**(1.3) | **74.8**(1.3) | **76.8**(1.2) | 59.5(3.7) | 65.3(1.9) | 67.8(2.1) |
| | *GPT-4* | | | | | | | | | |
| | + Rerank (L) | 60.8(2.3) | 62.6(2.7) | 63.0(1.3) | 65.9(2.7) | 72.3(0.3) | 74.5(1.5) | 59.6(2.9) | 64.9(2.5) | 67.1(2.5) |
| | + Ensemble (S) + Rerank (L) | 61.1(2.2) | 62.8(0.9) | 63.6(1.2) | 68.6(1.3) | 73.9(1.4) | 75.9(2.4) | **60.9**(3.9) | **65.6**(1.5) | **67.8**(1.7) |

Table 4: The F1-score differences before and after reranking on the reranked samples, as well as their proportion of the total samples.

| | GPT-4 | | | | InstructGPT | | | |
|---|---|---|---|---|---|---|---|---|
| | before | after | △ | ratio | before | after | △ | ratio |
| FewNER | 31.9 | 40.7 | 8.8 | 3.2% | 31.4 | 28.3 | −3.1 | 3.3% |
| TACREV | 25.3 | 43.0 | 17.7 | 9.1% | 33.8 | 43.4 | 9.6 | 7.1% |
| ACE05 | 31.1 | 57.9 | 26.8 | 1.6% | 35.6 | 55.7 | 20.1 | 0.5% |

Table 5: Ablation study on three datasets. The filter is ensembled SLMs and the reranker is GPT-4.

| CoT | Demo | LF | AD | FewNERD (20-shot) | TACREV (100-shot) | ACE05 (20-shot) |
|---|---|---|---|---|---|---|
| ✓ | ✓ | ✓ | ✓ | 63.6(1.2) | **75.9**(2.4) | **67.8**(1.7) |
| ✗ | ✓ | ✓ | ✓ | 63.2(1.2) | 75.4(2.4) | 67.2(1.7) |
| ✗ | ✗ | ✓ | ✓ | 63.0(1.4) | 74.9(2.2) | 66.6(1.5) |
| ✗ | ✗ | ✗ | ✓ | 62.4(2.1) | 73.8(2.5) | 66.5(1.3) |
| ✗ | ✗ | ✗ | ✗ | 12.5(2.7) | 59.9(6.0) | 5.4(1.1) |
| Previous SoTA methods | | | | 62.6(1.0) | 74.1(1.7) | 66.5(1.7) |

We remove all examples, rendering the reranking a zero-shot problem. (3) **LF** (label filtering): We retain all labels as candidate choices for reranking, instead of only the top-$N$ labels from the SLMs. (4) **AD** (adaptive): We feed all samples, not just hard ones, to the LLMs.

We show their results in Table 5 and see that (1) Demos with explanations consistently enhance the reranking ability of LLMs across all datasets. (2) Demos without explanations also contribute to performance improvement. (3) Label filtering results in gains and notably reduces the demo length,

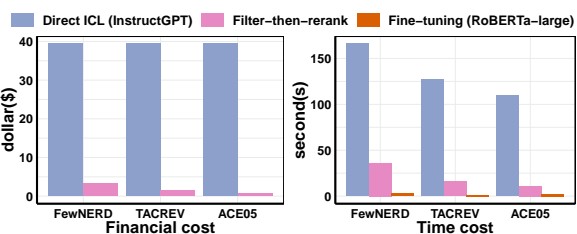

Figure 7: The financial and time cost over 500 sentences. InstructGPT as the reranker.

hence cutting inference costs. (4) The performance collapses without a filter to identify sample difficulty, reiterating the need for an integrated SLM-LLM system to complement each other.

## 6 Conclusion

Through an extensive empirical study on nine datasets spanning four IE tasks, we find that LLMs, despite their superiority in extreme low-resource scenarios, are not effective few-shot information extractors in general. They struggle with IE-related prompts, have limited demonstration capacity, and incur high inference costs. However, LLMs significantly improve the performance on *hard* samples when combined with SLM. Building on these insights, we propose an adaptive *filter-then-rerank* paradigm to leverage the strengths of SLMs and LLMs and mitigate their limitations. This approach consistently achieves promising results, with an average 2.4% F1 gain across multiple few-shot IE tasks, while minimizing latency and budget costs.

## Limitations

We do work hard to find better prompts to elicit the power of LLMs on few-shot IE tasks in Section 3.5, by exploring various kinds of LLMs, demonstration strategies and prompt formats. We find that different prompt variants do not significantly impact in-context learning abilities. As an empirical study, we acknowledge the potential existence of a *lottery* prompt superior to our explored prompts. However, it seems unlikely that an improved prompt would substantially alter our conclusions.

Another common risk when evaluating LLMs on public benchmark is their potential memorization of samples tested. To mitigate such potential contamination, we use earlier and stable versions of these models rather than the newer and updated ones (for example, `gpt-4-0314` instead of `gpt-4`). Even if such contamination makes abilities of LLMs overestimated, our primary conclusions remain unchanged because we find that LLMs are **NOT** good few-shot information extractors.

Regarding our adaptive *filter-then-rerank* paradigm, a key limitation lies in how to assess sample difficulty. In this work, we employ a simple unsupervised metric, *i.e.,* the maximum probabilities from SLMs. This is predicated on the assumption that SLMs are well-calibrated (Guo et al., 2017). However, it is an obviously imperfect assumption. We envision that calibrating SLMs-based filters or developing an advanced difficulty metric could substantially enhance LLM rerankers' performance. We leave them for future work.

## Acknowlegement

This study is supported under the RIE2020 Industry Alignment Fund – Industry Collaboration Projects (IAF-ICP) Funding Initiative, the Singapore Ministry of Education (MOE) Academic Research Fund (AcRF) Tier 1 grant, as well as cash and in-kind contribution from the industry partner(s).

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

## A  Datasets

### A.1  Full Datasets

We construct few-shot IE datasets and conduct the empirical study on nine datasets spanning four tasks, **with varying schema complexities ranging from 4 to 168**. We show their statistics in Table 6.

### A.2  Details of Few-shot IE Datasets

**Sampling Algorithm for Train/Valid Datasets.** We downsample sentences from original training dataset to construct few-shot training and valid datasets. We adopt $K$-shot sampling strategy that each label has (at least) $K$ samples. We set 6 $K$-values (1, 5, 10, 20, 50, 100) for RE tasks and 4 $K$-values (1, 5, 10, 20) for other tasks. For RE task, each sentence has exactly one relation and we simply select $K$ sentences for each label. For NER, ED and EAE tasks, each sentences is possible to contain more than one entities/events/arguments. Since our sampling is at sentence-level, the algorithm of accurate sampling , *i.e.,* finding exactly $K$ samples for each label, is NP-complete[8] and unlikely to find a practical solution. Therefore we follow Yang and Katiyar (2020) adopting a greedy sampling algorithm to select sentences for NER and ED tasks, as shown in Algorithm 1. Note that the actual sample number of each label can be larger than $K$ under this sampling strategy. For all three tasks, we additionally sample negative sentences (without any defined labels) and make the ratio of positive sentences (with at least one label) and negative sentences as 1:1. The statistics of the curated datasets are listed in Table 7.

---

[8]The *Subset Sum Problem*, a classical NP-complete problem, can be reduced to this sampling problem.

---

**Algorithm 1** Greedy Sampling

**Require:** shot number $K$, original full dataset $\mathcal{D} = \{(\mathbf{X}, \mathbf{Y})\}$ tagged with label set $E$
1: Sort $E$ based on their frequencies in $\{\mathbf{Y}\}$ as an ascending order
2: $S \leftarrow \phi$, Counter $\leftarrow$ dict()
3: **for** $y \in E$ **do**
4:     Counter$(y) \leftarrow 0$
5: **end for**
6: **for** $y \in E$ **do**
7:     **while** Counter$(y) < K$ **do**
8:         Sample $(\mathbf{X}, \mathbf{Y}) \in \mathcal{D}$ s.t.$\exists j, y_j = y$
9:         $\mathcal{D} \leftarrow \mathcal{D} \backslash (\mathbf{X}, \mathbf{Y})$
10:         Update Counter (not only $y$ but all event types in $\mathbf{Y}$)
11:     **end while**
12: **end for**
13: **for** $s \in \mathcal{S}$ **do**
14:     $\mathcal{S} \leftarrow \mathcal{S} \backslash s$ and update Counter
15:     **if** $\exists y \in E$, s.t. Counter$(y) < K$ **then**
16:         $\mathcal{S} \leftarrow \mathcal{S} \bigcup s$
17:     **end if**
18: **end for**
19: **return** $\mathcal{S}$

---

Based on the subsets constructed above, we optionally further split them into training and valid sets. For few-shot datasets with more than 300 sentences, we additionally split 10% sentences as the valid set and the remaining sentences as training set. Otherwise, we do not construct valid set and conduct 5-fold cross validation to avoid overfitting.

## B  Details on SLMs

We adopt five representative supervised methods to evaluate the ability of SLMs on few-shot IE tasks.
**(1). Fine-tuning (FT)**: Add a classifier head on SLMs to predict the labels of each sentence/word.
**(2). FSLS** (Ma et al., 2022a): The state-of-the-art extractive-based method for few-shot NER task. Ma et al. (2023) also validate its competitive performance on few-shot ED tasks.
**(3). KnowPrompt** (Chen et al., 2022b): The best extractive-based method for few-shot RE task.
**(4). PAIE** (Ma et al., 2022b): The best extractive-based method for few-shot EAE task.
**(5). UIE** (Lu et al., 2022c): A competitive unified generation-based method for few-shot IE tasks. We introduce their implementation details below:
**Fine-tuning/FSLS.** We implement these two meth-

Table 6: Statistics of nine datasets used. Note that the #*mentions* for event detection tasks refers to the number of trigger words, while the #*mentions* for event argument extraction tasks refers to the number of arguments.

| | | Named Entity Recognition | | | Relation Extraction | | Event Detection | | | Event Arg Extraction | | |
| Dataset | | CONLL | OntoNotes | FewNERD | TACREV | TACRED | ACE05 | MAVEN | ERE | ACE05 | RAMS | ERE |
|---|---|---|---|---|---|---|---|---|---|---|---|---|
| #Label Type | | 4 | 18 | 66 | 41 | 41 | 33 | 168 | 38 | 33 | 139 | 38 |
| #Sents | Train | 14,041 | 49,706 | 131,965 | 68,124 | 68,124 | 14,024 | 32,360 | 14,736 | 14,024 | 7329 | 14,736 |
| | Test | 3,453 | 10,348 | 37,648 | 15,509 | 15,509 | 728 | 8,035 | 1,163 | 728 | 871 | 1,163 |
| #Mentions | Train | 23,499 | 128,738 | 340,247 | 13,012 | 13,012 | 5,349 | 77,993 | 6,208 | 4859 | 17026 | 8924 |
| | Test | 5,648 | 12,586 | 96,902 | 3,123 | 3,123 | 424 | 18,904 | 551 | 576 | 2023 | 822 |

Table 7: The statistics of few-shot training sets. We set different random seeds and generate 5 training sets for each setting. We report their average statistics.

| Dataset Settings | | # Labels | # Sent | # Sample | # Avg shot |
|---|---|---|---|---|---|
| CONLL'03 | 1-shot | 4 | 4.8 | 5.8 | 1.4 |
| | 5-shot | | 16.2 | 21.8 | 5.5 |
| | 10-shot | | 29.2 | 42.6 | 10.7 |
| | 20-shot | | 65.6 | 82.0 | 20.5 |
| OntoNotes | 1-shot | 18 | 20.0 | 33.4 | 1.9 |
| | 5-shot | | 84.8 | 148.0 | 8.2 |
| | 10-shot | | 158.6 | 281.0 | 15.6 |
| | 20-shot | | 332.8 | 547.2 | 30.4 |
| FewNERD | 1-shot | 66 | 89.8 | 147.0 | 2.2 |
| | 5-shot | | 286.2 | 494.8 | 7.5 |
| | 10-shot | | 538.0 | 962.0 | 14.6 |
| | 20-shot | | 1027.2 | 1851.4 | 28.1 |
| TACREV | 1-shot | 41 | 81.6 | 41.0 | 1.0 |
| | 5-shot | | 387.6 | 205.0 | 5.0 |
| | 10-shot | | 741.2 | 406.0 | 9.9 |
| | 20-shot | | 1367.2 | 806.0 | 19.7 |
| | 50-shot | | 2872.0 | 1944.0 | 47.4 |
| | 100-shot | | 4561.0 | 3520.0 | 85.9 |
| TACRED | 1-shot | 41 | 81.6 | 41.0 | 1.0 |
| | 5-shot | | 387.6 | 205.0 | 5.0 |
| | 10-shot | | 741.2 | 406.0 | 9.9 |
| | 20-shot | | 1367.2 | 806.0 | 19.7 |
| | 50-shot | | 2871.2 | 1944.0 | 47.4 |
| | 100-shot | | 4575.2 | 3520.0 | 85.9 |
| ACE05 | 1-shot | 33 | 47.4 | 41.0 | 1.2 |
| | 5-shot | | 192.8 | 165.0 | 5.0 |
| | 10-shot | | 334.6 | 319.4 | 9.7 |
| | 20-shot | | 579.4 | 598.2 | 18.1 |
| MAVEN | 1-shot | 168 | 157.6 | 298.0 | 1.8 |
| | 5-shot | | 540.4 | 1262.2 | 7.5 |
| | 10-shot | | 891.2 | 2413.8 | 14.4 |
| | 20-shot | | 1286.4 | 4611.4 | 27.4 |
| ERE | 1-shot | 38 | 48.4 | 54.6 | 1.4 |
| | 5-shot | | 175.0 | 219.2 | 5.8 |
| | 10-shot | | 304.8 | 432.4 | 11.4 |
| | 20-shot | | 521.6 | 806.6 | 21.2 |
| ACE05 | 1-shot | 33 | 23.4 | 40.2 | 1.2 |
| | 5-shot | | 79.8 | 178.2 | 5.4 |
| | 10-shot | | 130.8 | 337.4 | 10.2 |
| | 20-shot | | 213.4 | 630.2 | 19.1 |
| RAMS | 1-shot | 139 | 130.2 | 332.6 | 2.4 |
| | 5-shot | | 514.0 | 1599.6 | 11.5 |
| | 10-shot | | 795.2 | 3193.2 | 23.0 |
| | 20-shot | | 1070.4 | 6095.4 | 43.9 |
| ERE | 1-shot | 38 | 21.6 | 102.8 | 2.7 |
| | 5-shot | | 74.2 | 403.4 | 10.6 |
| | 10-shot | | 127.2 | 775.6 | 20.4 |
| | 20-shot | | 190.2 | 1397.2 | 36.8 |

ods by ourselves. We use RoBERTa-large (Liu et al., 2019) as the backbones. We adopt Automatic Mixed Precision (AMP) training strategy[9] to save memory. We run each experiment on a single NVIDIA V100 GPU. We train each model with the AdamW (Loshchilov and Hutter, 2019) optimizer with linear scheduler and 0.1 warm-up steps. We set the weight-decay coefficient as 1e-5 and maximum gradient norms as 1.0. We set the batch size as 64, the maximum input length as 192, the training step as 500 and the learning rate as 5e-5.

**KnowPrompt** We implement this method based on original source code[10], and use RoBERTa-large as our backbones. We set 10 maximum epochs for 50- and 100-shot datasets, and as 50 epochs for other datasets. We keep all other hyperparameters as default, and run each experiment on a single NVIDIA V100 GPU.

**PAIE** We implement this method on original source code[11], and use BART-large (Lewis et al., 2020) as backbones. We keep all hyperparameters as default for ACE and RAMS dataset. For ERE dataset, we set the training step as 1000, the batch size as 16 and the learning rate as 2e-5. We run each experiment on a single NVIDIA V100 GPU.

**UIE** We implement this method based on original source code[12], and use T5-large (Raffel et al., 2020) as the backbones. We run each experiment on a single NVIDIA Quadro RTX8000 GPU. We set the batch size as 4 with 4000 training steps. We set the maximum input length as 800 and the learning rate as 1e-4.

## C  LLMs Implementations

Regarding our empirical study, we explore the ICL abilities of LLMs on few-shot IE tasks. We mainly use five LLMs from two sources. (1) OpenAI

[9]https://pytorch.org/docs/stable/amp.html
[10]https://github.com/zjunlp/KnowPrompt
[11]https://github.com/mayubo2333/PAIE
[12]https://github.com/universal-ie/UIE

models: CODEX (`code-davinci-002`; Chen et al. 2021), InstructGPT (`text-davinci-003`; Ouyang et al. 2022), and ChatGPT (`gpt-3.5-turbo-0301`). (2) Open-source models: LLaMA-13B (Touvron et al., 2023) and its instruction-tuned counterpart, Vicuna-13B (Chiang et al., 2023). We detail their implementation details in the next sections below.

## C.1 Open-source Models

We implement multiple ICL approaches on LLaMA-13B and Vicuna-13B without fine-tuning. We set the maximum input length as 2048 and the batch size as 1. We run each experiment on a single NVIDIA V100 GPU. To achieve this, we leverage the `Accelerate` [13] framework and fp16 inference to save memory. We set maximum output length as 96 and sampling temperature as 0 (*i.e.,* greedy decoding). We set both `frequency_penalty` and `presence_penalty` as 0.

## C.2 OpenAI Models

We implement multiple ICL approaches on OpenAI models by calling their official APIs [14]. We set the maximum input length as 3600 for all tasks and models. The only exception occurs when we use CODEX on RE tasks, where we set the maximum input length as 7000. We unify the maximum output length as 32 for RE task, and 96 for other three tasks. We set the sampling temperature coefficient as 0, *i.e.,* greedy decoding.

## D Pivot Experiments on LLMs

### D.1 Sampling Temperature

Existing prompt-engineering discussion[15] suggests setting the sampling temperature $t = 0$ for tasks with structured outputs, including IE tasks. We validate this conclusion in Table 8, from which we could see the generated quality when $t = 0$ is much higher than the quality when $t \neq 0$. Therefore we set $t = 0$ in all main experiments, and do not take self-consistency (Wang et al., 2023c) into account.

### D.2 Automatic Chain-of-thought

We additionally investigate whether rationales could facilitate LLMs' performance on few-shot IE tasks. Since there exists no golden rationales in

---

[13]https://huggingface.co/docs/accelerate
[14]https://openai.com/blog/openai-api
[15]https://help.openai.com/en/articles/6654000-best-practices-for-prompt-engineering-with-openai-api

Table 8: F1-scores across different $t$ values. Experiments run on 10-shot settings with CODEX.

| | FewNERD | TACREV | ACE05 |
|---|---|---|---|
| $t = 0$ | $48.5_{(1.9)}$ | $53.7_{(2.3)}$ | $42.9_{(2.2)}$ |
| + 5-ensemble | $\mathbf{53.5}_{(1.3)}$ | $\mathbf{58.6}_{(1.5)}$ | $\mathbf{46.3}_{(0.8)}$ |
| $t = 0.7$ | $40.9_{(2.3)}$ | $39.9_{(1.2)}$ | $35.6_{(1.0)}$ |
| + self-consistency | $52.1_{(0.9)}$ | $53.4_{(1.3)}$ | $45.6_{(3.0)}$ |

original datasets, we follow Automatic Chain-of-thought (Auto-CoT; Zhang et al. 2023b) method as below. Regarding each sample, we query LLMs

*According to [sentence], Why [span] is a [label].*

For example, given the sentence *"DSC and Traction Control on all Speed3 models is also standard."*, we would feed LLM the query that *"Could you explain why Speed3 is a kind of car"*. Then we insert the bootstrapped rationales between the sentences and ground-truth answers. If a sentence has no positive labels, however, we do not ask LLMs and keep the original format as the vanilla ICL approach. Here we prompt InstructGPT to generate the rationales with temperature $t = 0.7$. We compare the performance with and without Auto-CoT as shown in Table 9.

Table 9: The F1-score difference between with and without Auto-CoT. We generate rationales by InstructGPT, then adopt **ICL w. Auto-CoT** approach and use CODEX as our backbone for inference.

| 10-shot train set | FewNERD (NER) | TACREV (RE) | ACE05 (ED) |
|---|---|---|---|
| wo. Auto-CoT | $\mathbf{54.0}_{(1.4)}$ | $\mathbf{57.3}_{(1.8)}$ | $\mathbf{47.7}_{(2.8)}$ |
| w. Auto-CoT | $36.6_{(1.7)}$ | $22.0_{(1.2)}$ | $43.1_{(3.4)}$ |

We are frustrated to find Auto-CoT degrades the performance with a large margin. We speculate this degration could be attributed to three main reasons. (1) The rationale increase the length of each sample and thus decrease the overall example number in demos. (2) There exists an obvious discrepancy between sentences with and without positive labels. The rationales are only provided for sentences with positive labels because it is hard to explain why a sentence dose not contain any label. (3) Some auto-generated rationales are low-quality, especially for RE tasks. We would explore better strategy to exploit auto-genertaed rationales in the future work.

Table 10: F1-scores difference among GPT-4, CODEX and InstructGPT.

| | NER (20-shot) | | | RE (100-shot) | | ED (20-shot) | | | EAE (20-shot) | | |
| | CONLL | OntoNotes | FewNERD | TACREV | TACRED | ACE05 | MAVEN | ERE | ACE05 | RAMS | ERE |
|---|---|---|---|---|---|---|---|---|---|---|---|
| InstructGPT | 77.2 | 47.7 | 57.2 | **62.7** | **53.8** | 49.3 | 25.4 | **40.8** | **45.8** | **42.2** | **41.9** |
| CODEX | 81.1 | 55.6 | 55.9 | 62.4 | 53.6 | 47.9 | 22.8 | 39.0 | - | - | - |
| GPT-4 | **84.7** | **65.6** | **57.8** | 59.3 | 50.4 | **52.1** | **30.2** | 40.5 | 42.9 | 38.6 | 38.2 |
| Supervised SoTA | 72.3 | 74.9 | 61.4 | 72.6 | 63.1 | 65.8 | 54.7 | 56.2 | 55.2 | 57.7 | 55.6 |

## D.3 GPT-4 v.s. Others

We tend to minimize the GPT-4 calls due to its high price. Thus we utilize 20-/100-shot settings across each dataset to compare GPT-4's performance with other LLMs. Table 10 reveals that GPT-4 does not outperform other LLMs significantly, except on OntoNotes and MAVEN. However, even on these datasets, GPT-4 still falls behind supervised SLMs by a significant margin. Consequently, the exclusion of GPT-4 does not undermine the conclusions drawn from our main experiments, and we omit it from our empirical study.

## E  Auxiliary Experiments

### E.1  LLMs struggle on Fine-grained Datasets

Based on the results shown in Figure 2, we additionally provide a quantitative analysis to show that LLMs struggle with fine-grained datasets. Under the 5-shot setting, we compare the performance difference of LLMs (ChatGPT) and SLMs (SoTA few-shot models) among different datasets. For each IE task, we observe a clear negative corre-

Table 11: Performance comparison between LLMs (ChatGPT) and SLM-based methods among datasets with various schema complexities.

| Named Entity Recognition | | | |
|---|---|---|---|
| | CoNLL | OntoNotes | FewNERD |
| # Entity | 4 | 18 | 66 |
| Micro-F1 (SLM) | 52.5 | 59.7 | 59.4 |
| Micro-F1 (LLM) | 77.8 | 59.4 | 55.5 |
| $\Delta$F1 (LLM, SLM) | 25.3 | -0.3 | -3.9 |

| Event Detection | | | |
|---|---|---|---|
| | ACE05 | ERE | MAVEN |
| # Event | 33 | 38 | 168 |
| Micro-F1 (SLM) | 55.1 | 48.0 | 49.4 |
| Micro-F1 (LLM) | 39.6 | 33.8 | 25.3 |
| $\Delta$F1 (LLM, SLM) | -15.5 | -14.2 | -24.1 |

| Event Argument Extraction | | | |
|---|---|---|---|
| | ACE05 | ERE | RAMS |
| # Event / #Role | 33 / 22 | 38 / 26 | 139 / 65 |
| Head-F1 (SLM) | 45.9 | 40.4 | 54.1 |
| Head-F1 (LLM) | 52.8 | 40.7 | 44.2 |
| $\Delta$F1 (LLM, SLM) | 6.9 | 0.3 | -9.9 |

lation between the label number (row 2) and the performance difference (row 5). In other words, with more label types, LLMs tend to perform relatively worse than SLMs. Therefore we conclude that LLMs struggle on fine-grained datasets.

### E.2  Finding Better Instruction

To investigate whether LLMs would benefit from complex instructions, we explored six instruction variants from simple to complex. Take NER task as an example, we illustrate them as below.

**Instruction0**: `[empty]`

**Instruction1**: `Identify the entities expressed by each sentence, and locate each entity to words in the sentence. The possible entity types are: [Type_1], [Type_2], ..., [Type_N]. If you do not find any entity in this sentence, just output 'Answer: No entities found.'`

**Instruction2**: `Identify the entities expressed by each sentence, and locate each entity to words in the sentence. The possible entity types are:`

- `[Type_1]: [Definition_1]`
- `[Type_2]: [Definition_2]`
- `...`
- `[Type_N]: [Definition_N]`

`If you do not find any entity in this sentence, just output 'Answer: No entities found.'`

**Instruction3**: `Assume you are an entity-instance annotator. Given a sentence, you need to (1) identify the word or phrase about the entity in the sentence, and (2) classify its entity type. The possible entity types are listed as below: [Type_1], [Type_2], ..., [Type_N]. Please note that your annotation results must follow such format: "'Answer: ([Type_1] <SEP>`

identified_entity:[Entity_1]), ([Type_2] <SEP> identified_entity:[Entity_2]), ......"'. If you do not find any entity in this sentence, just output 'Answer: No entities found.'

**Instruction4**: Assume you are an entity-instance annotator. Your objective is to perform a series of intricate steps for Named Entity Recognition. Firstly, you have to identify a particular word or phrase in the sentence that corresponds to an entity. Following this, classify the entity into one of the potential entity types. The potential entity types are provided as below: [Type_1], [Type_2], ..., [Type_N]. Please note that your annotation results must follow such format: 'Answer: ([Type_1] <SEP> identified_entity:[Entity_1]), ([Type_2] <SEP> identified_entity:[Entity_2]), ......'. If you do not find any entity in this sentence, just output 'Answer: No entities found.'

**Instruction5**: Assume you are an entity-instance annotator. Given a sentence, you need to (1) identify the word or phrase about the entity in the sentence, and (2) classify its entity type. The possible entity types are listed as below:

- [Type_1]: [Definition_1]

- [Type_2]: [Definition_2]

- ...

- [Type_N]: [Definition_N]

Please note that your annotation results must follow such format: 'Answer: ([Type_1] <SEP> identified_entity:[Entity_1]), ([Type_2] <SEP> identified_entity:[Entity_2]), ......'. If you do not find any entity in this sentence, just output 'Answer: No entities found.'

Regarding these six instructions, we evaluate their performance of ChatGPT on four 20-shot IE tasks. As shown in Table 12, there is no significant correlation between the instruction complexity

Table 12: F1-scores across six instruction formats. Experiments run on 20-shot settings with ChatGPT.

|    | **FewNERD** (NER) | **TACREV** (RE) | **ACE** (ED) | **ACE** (EAE) |
|----|-------------------|-----------------|--------------|---------------|
| I0 | $57.6_{(2.1)}$ | $49.1_{(2.4)}$ | $44.0_{(1.4)}$ | $50.9_{(0.1)}$ |
| I1 | $58.3_{(0.5)}$ | $49.6_{(1.2)}$ | $42.6_{(1.0)}$ | $51.5_{(1.1)}$ |
| I2 | $57.7_{(1.0)}$ | $50.0_{(1.5)}$ | $41.8_{(0.9)}$ | $50.3_{(1.5)}$ |
| I3 | $57.6_{(2.3)}$ | $52.3_{(1.8)}$ | $42.9_{(1.3)}$ | $49.2_{(2.3)}$ |
| I4 | $56.8_{(0.9)}$ | $49.6_{(2.9)}$ | $41.6_{(1.9)}$ | $49.9_{(1.2)}$ |
| I5 | $57.8_{(0.5)}$ | $47.2_{(1.8)}$ | $43.1_{(1.8)}$ | $50.6_{(1.8)}$ |

and LLMs' performance. Even the prompt without instruction (I0) leads to comparable, if not better, results than prompt with complex instructions. Therefore, we use simple instruction (I1) in our main experiment.

### E.3 Do More Samples in Demos Help?

We wonder whether longer demos bring more powerful ICL abilities for LLMs. Thus we investigate the impact of increasing the number of demonstrations on LLMs' performance in Figure 8. We observe that: (1) The performance of the RE task consistently improves with more demos, indicating its potential benefiting from additional annotations. (2) The NER and ED tasks reach a stable or degraded performance with increased demo numbers, suggesting that they are limited even before reaching the maximum input length. (3) Open-source LLMs, *i.e.,* LLaMA and Vicuna, have more limited capacities in leveraging demos compared to OpenAI models, with their performance stagnating or even collapsing with only a few (2-4) demos.

### E.4 Finding Better Demo Selection Strategy

The maximum input length of LLMs usually limits the sentence number in demos even under few-shot settings. For each test sentence $s$, we demand a demo retriever $\mathcal{E}(D, s)$ which selects a subset from $D$ as the sentences in demo. Following previous work, we consider three commonly-used strategies. (1) Random sampling. (2) Sentence-embedding (Liu et al., 2022; Su et al., 2022): retrieving the top-K nearest sentences measured by sentence embedding. We compute the embeddings by SimCSE-RoBERTa-large (Gao et al., 2021).

$$\mathcal{E}(D, s) = \text{arg-topK}_{s' \in D}[\text{Sent-embed}(s', s)] \quad (3)$$

(3) Efficient Prompt Retriever (Rubin et al., 2022): retrieving by a neural retriever $R$ trained on $D$.

$$\mathcal{E}(D, s) = \text{arg-topK}_{s' \in D}[R_D(s', s)] \quad (4)$$

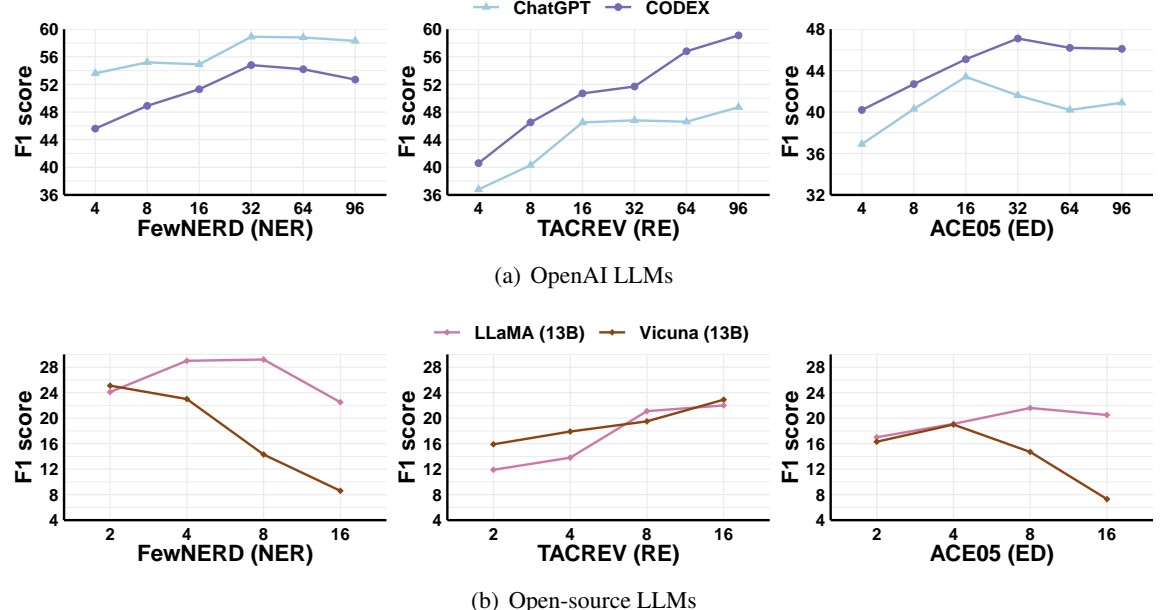

(a) OpenAI LLMs

(b) Open-source LLMs

Figure 8: Relationship between demo number and F1-score among three datasets. Note that the x-axis in each subfigure represents the number of demos (not the shot value $K$) during ICL. We adopt sentence embedding as the demo selection strategy and text prompt in this experiment.

For each test sentence $s$, we pre-retrieve $M$ similar sentences $\bar{D} = \{(s'_i, y'_i)\}_{i=1}^{M} \subset D$. Then we score each sentence in $\bar{D}$ by their likelihoods $P_{\mathcal{L}}(f(y'_i)|f(s'_i))$ where $f$ denotes the prompt format adopted and $\mathcal{L}$ the scoring LM. We randomly select positive samples $s'^{(\text{pos})}_i$ from the top-$K_D$ sentences and hard negative samples $s'^{(\text{hard-neg})}_i$ from the bottom-$K_D$ ones. Then we train $R_D$ by in-batch contrastive learning (Chen et al., 2020). For each sentence $s'_i$ within the batch, there are 1 positive sentences $s'^{(\text{pos})}_i$ and $2B-1$ negative sentences $\{s'^{(\text{hard-neg})}_j\}_{j=1}^{B} \cup \{s'_j\}_{j\neq i}^{B}$. Here we adopt $M$ as 40, $K_D$ as 5, $f$ as text prompt, the batch size $B$ as 128, and the scoring LM $\mathcal{L}$ as FLAN-T5-xl.

Table 13: F1-scores on three demo-selection strategies. Experiments run on 20-shot settings with ChatGPT.

|  | FewNERD (NER) | TACREV (RE) | ACE (ED) |
|---|---|---|---|
| Random Sampling | 53.2$_{(0.4)}$ | 43.0$_{(3.3)}$ | 38.0$_{(1.5)}$ |
| Sentence Embedding | **57.6**$_{(2.3)}$ | **49.6**$_{(1.2)}$ | 42.9$_{(1.3)}$ |
| Efficient Prompt Retriever | 57.2$_{(0.6)}$ | 48.0$_{(0.8)}$ | **43.5**$_{(1.4)}$ |

Table 13 demonstrates the F1-score performance on different selection strategies. We find that both the sentence embedding and EPR surpass random sampling by a large margin. Given the simplicity of the sentence embedding, we adopt it, rather than EPR, as our selection strategy in main experiment.

Table 14: F1-scores across three prompt formats. Experiments run on 20-shot settings with ChatGPT.

|  | FewNERD (NER) | TACREV (RE) | ACE (ED) | ACE (EAE) |
|---|---|---|---|---|
| Text | 57.6$_{(2.3)}$ | 49.6$_{(1.2)}$ | 42.9$_{(1.3)}$ | **51.5**$_{(1.1)}$ |
| Code | 53.2$_{(0.9)}$ | **50.2**$_{(1.8)}$ | **44.3**$_{(2.0)}$ | 47.3$_{(1.5)}$ |

### E.5 Finding Better Prompt Format

Previous studies on LLMs for few-shot IE tasks have explored different prompt formats and highlighted the importance of selecting an appropriate format for achieving competitive performance. Therefore, we investigate two commonly-used variants in previous work: (1) Text prompt as shown in Figure 1. (2) Code prompt: We follow Wang et al. (2023b); Li et al. (2023) and recast the output of IE tasks in the form of code. See more details about this format in their original papers.

Table 14 shows comparable performance across all formats. Based on simplicity, we choose the text prompt for our main experiment.

## F  Case Study

### F.1  Hard Samples

Table 15 showcases some *hard* examples which benefits from our LLM reranking. In accordance with our intuition, we observe that the LLM

rerankers correct two kinds of erroneous predictions made by LLMs. (1) The lack of external knowledge, such as the first (*Triptolemus is a figure in Greek mythology*) and third examples (*Minas Gerais is a state instead of city*). (2) Limited reasoning abilities, such as the second (*His wife's children are his children*) and the fourth (*The word "fought" in this sentence does not involve any physical violence*) examples.

### F.2 Easy Samples

As discussed in Section 4.3, we attribute the inferior performance of LLMs on easy samples to their false-positive predictions for negative samples. Upon examining the incorrect cases, we speculate that two main reasons are responsible: (1) LLM hallucination and (2) span boundary mismatch. We provide details on them with specific cases below.

**LLM Hallucination**. The most distinctive mistakes made by LLMs. Without specific fine-tuning on IE tasks, LLMs (with in-context learning approaches) sometimes hallucinate and provide meaningless, even confusing explanations or answers. For instance, LLMs might predict a verb, preposition or even special symbols as an entity.

Case 1: LLMs predict the word *die* as entity person-other (Ground truth: None)

---

**Sentence:** Flavius Rumoridus (died 5th century AD) was a Roman soldier who was appointed consul in AD 403 in the Western Roman Empire at the same time as the eastern emperor Theodosius II served in the same capacity in the East.

**Candidate Choices:**

(a) died do/does not belong to any known entities.

(b) died is a kind of geopolitical entity.

(c) died is a person, but not affiliated with following professions: actor, artist, athlete, author, director, politician, scholar, soldier.

---

**Analysis:** The sentence provides information about Flavius Rumoridus, who was a Roman soldier and appointed consul in AD 403. The sentence also mentions that he died in the 5th century AD, which is a statement about his death. Therefore, the correct answer is (c) because the sentence does not provide any information about Flavius Rumoridus being affiliated with any of the professions listed in option (c).

**Correct Answer:** (c)

---

Case 2: LLMs predict the word *of* as entity organization-other (Ground truth: None)

---

**Sentence:** During these years he studied at the South Kensington Schools of the Royal College of Art.

**Candidate Choices:**

(a) of do/does not belong to any known entities.

(b) of is an educational institute.

(c) of pertains to an organization that does not fall under the categories of company, educational institution, government, media, political party, religion, sports league, sports team, band or musical group.

**Analysis:** In the given sentence, *of* is a preposition that indicates the relationship between *South Kensington Schools* and *Royal College of Art*. None of the candidate choices directly address the correct relationship between these entities or state that they are not applicable. Therefore, the correct answer is (c): *of* does not pertain to any known entities mentioned in the choices provided.

**Correct Answer:** (c)

---

**Span boundary mismatch**. Different from sentence-level approaches which take one or multiple sentences as an unit and assign each word's label globally, LLMs under our filter-then-rerank

paradigm adopt sample-level approaches. In other words, LLMs only consider each word/span candidate independently. The sample-level approaches enable LLMs to rerank candidates with the help of multi-choice question prompt. However, they also aggravate the span boundary mismatching problem in NER and ED tasks. We empirically find easy samples suffering much more from boundary mismatching problem than hard samples.

Case 3: LLMs predict *Baron*, a subspan of *Baron Carl Hurleman*, as the entity `person-other` (The ground-truth entity type of *Baron* is None. And the ground-truth entity type of *Baron Carl Hurleman* is `person-artist/author`).

---

**Sentence:** Additions in the 1740s were by Baron Carl Hurleman (1700 - 1753).

**Candidate Choices:**

(a) Baron do/does not belong to any known entities.

(b) Baron is a person, but not affiliated with following professions: actor, artist, athlete, author, director, politician, scholar, soldier.

(c) Baron is a kind of geopolitical entity

**Analysis:** In the given sentence, *Baron* is a title given to a person named Carl Hurleman, who is mentioned as the one who made additions in the 1740s. The sentence does not provide any information about connections to geopolitical entities or any other known entities mentioned in the choices provided. Therefore, the correct answer is (b): Baron is a person, but not affiliated with following professions: actor, artist, athlete, author, director, politician, scholar, soldier.

**Correct Answer:** (b)

---

Case 4: LLMs predict *65th*, a subspan of *65th British Academy Film Awards*, as the entity `other-awards` (The ground-truth entity type of *65th* is None. And the ground-truth entity type of *65th British Academy Film Awards* is `other-awards`).

---

**Sentence:** Ralph Fiennes was nominated for the BAFTA Award for Outstanding Debut by a British Writer, Director or Producer at the 65th British Academy Film Awards.

**Candidate Choices:**

(a) 65th do/does not belong to any known entities.

(b) 65th is a kind of award.

(c) 65th refers to some event except attack, election, natural disaster, protest, revolution and sports.

**Analysis:** The sentence mentions the *65th British Academy Film Awards*, which is an event that recognizes outstanding achievements in the film industry. Therefore, option (c) is incorrect as it excludes events like award ceremonies. Option (a) is also incorrect as the number *65th* is associated with the event. Hence, the correct answer is (b) as *65th* refers to the specific edition of the BAFTA Awards.

**Correct Answer:** (b)

---

# G Details on Adaptive *Filter-then-rerank* Paradigm

## G.1 Demonstration Examples

We convert few-shot IE tasks to multiple-choice questions in our *filter-then-rerank* paradigm. We show 4 examples used in demonstrations for FewNERD dataset in Table 16, for TACREV dataset in Table 17, and for ACE05 datasets in Table 18.

## G.2 Template

In our *filter-then-rerank* paradigm, we utilize templates converting candidate labels to question options. We list the template for FewNERD dataset in Table 19, for TACREV dataset in Table 20, and for ACE05 datasets in Table 21.

Table 15: Examples of the samples corrected by LLM reranking. We sample four examples from NER, RE and ED tasks, respectively. **Sentences**: The sentences in which samples locate. We color the samples (entities or trigger words) to be identified. **Before**: The prediction before LLM reranking. Based on SLM-based methods. **After**: The reranked prediction using LLMs. **Rationales**: LLM-generated Explanations.

| Sentences (entity/event) | Before | After | Rationales |
|---|---|---|---|
| **Named Entity Recognition** | | | |
| According to it, King Eleusis and Cothonea (Cyntinia), parents of Triptolemus, are visited by Demeter, who rears their son, feeding him divine milk by day and placing him into the fire at night, which makes Triptolemus grow faster than mortal children normally do. | person-other ✗ | other-god ✓ | According to the context, Triptolemus is a figure in Greek mythology, the son of King Eleusis and Cothonea (Cyntinia). He was a demigod who was sent by Demeter to teach mankind the art of agriculture. So Triptolemus is a god in some legend/religious story. |
| **Relation Extraction** | | | |
| Survivors include his wife, Sandra; four sons, Jeff, James, Douglas and Harris; a daughter, Leslie; his mother, Sally; and two brothers, Guy and Paul. | None ✗ | per:children ✓ | The sentence states 'Survivors include his wife, Sandra; four sons, Jeff, James, Douglas and Harris; a daughter, Leslie; his mother, Sally; and two brothers, Guy and Paul.' This indicates 'his' is the father of Jeff, James, Douglas and Harris, and the husband of Sandra. So his is the parent of Jeff. |
| Eliza Samudio, Bruno 's ex-girlfriend and his 4-month-old baby boy's mother, disappeared from her hometown of Belo Horizonte, Minas Gerais. | per:cities_of _residence ✗ | per:states_of _residence ✓ | The sentence states that 'Eliza Samudio, Bruno's ex-girlfriend and his 4-month-old baby boy's mother, disappeared from her hometown of Belo Horizonte, Minas Gerais.' This indicates that Eliza Samudio is from the city of Belo Horizonte, which is located in the state of Minas Gerais. So Eliza Samudio lives in the state or province Minas Gerais. |
| **Event Detection** | | | |
| We have fought every institution, every law firm ... it 's cost us a fortune but now we 're going on a massive expansion. | Conflict.Attack ✗ | None ✓ | In the given sentence, the word fought is used to describe the action of the speaker fighting against various institutions and law firms. This does not involve any physical violence or court proceedings, so the word fought does not trigger any known event. |

Table 16: Demo examples used in FewNERD dataset. We color the entity in blue.

---

**Instruct**: Read following sentences and identify what is the entity type of 392 quoted by <t>.
**Sentence**: Powell v. Texas , <t> 392 <t> U.S. 514 ( 1968 ) , was a United States Supreme Court case that ruled that a Texas statute criminalizing public intoxication did not violate the Eighth Amendment protection against cruel and unusual punishment.
(a) 392 is a legal document, a term or a convention in legal sense.
(b) 392 does not belong to any known entities.
(c) 392 refers to a protest, uprising or revolution event
(d) 392 refers to a government or governmental agency
**Analysis**: In the context you provided, 392 refers to the volume number in the United States Reports where the Supreme Court's decision in Powell v. Texas can be found. However, 392 itself does not refer to a legal document. So 392 do/does not belong to any known entities.
**Answer**: (b)

---

**Instruct**: Read following sentences and identify what is the entity type of The New Yorker quoted by <t>.
**Sentence**: In 2004 Gourevitch was assigned to cover the 2004 U.S. presidential election for " <t> The New Yorker <t> ".
(a) The New Yorker does not belong to any known entities.
(b) The New Yorker is a broadcast program.
(c) The New Yorker is a kind of written art.
(d) The New Yorker is a media/newspaper organization.
**Analysis**: The New Yorker is a well-known American magazine that has been published since 1925, and is primarily known for its long-form journalism, commentary, and satire. It has a reputation for publishing high-quality writing on a wide variety of topics, including politics, culture, and the arts. So The New Yorker is a media/newspaper organization.
**Answer**: (d)

---

**Instruct**: Read following sentence and identify what is the entity type of St. quoted by <t>.
**Sentence**: The May 1980 eruption of Mount <t> St. <t> Helens in the state of Washington seriously affected both 47th Air Division and 92d Bombardment Wing operations at Fairchild AFB , resulting in dispersal of Fairchild 's B-52 and KC-135 aircraft to various bases while around-the-clock shifts removed the volcanic ash from facilities within the base perimeter. "
(a) St. does not belong to any known entities.
(b) St. is a natural disaster event.
(c) St. is a geographic position about mountain.
**Analysis**: According to the context, St. is an abbreviation of Saint, used in the name of Mount St. Helens, which is an active volcano in the state of Washington. However, St. itself does not refer to anything. So St. do/does not belong to any known entities.
**Answer**: (a)

---

**Instruct**: Read following sentence and identify what is the entity type of Ridzuan quoted by <t>.
**Sentence**: <t> Ridzuan <t> was promoted to Harimau Muda A for 2014 season .
(a) Ridzuan does not belong to any known entities.
(b) Ridzuan is a person, but not affiliated with following professions: actor, artist, author, director, politician, scholar, soldier.
(c) Ridzuan is an athlete.
**Analysis**: The mention of 'Harimau Muda A' indicates that it is a sports-related context, and 'promoted' implies a progression or advancement within the sports team. So Ridzuan is an athlete.
**Answer**: (c)

Table 17: Demo examples used in TACREV dataset. We color the subject and object entities in blue.

**Instruct**: Read the sentence and determine the relation between she and lawyer quoted by <t>.
**Sentence**: The <t> lawyer <t> denied Italian news reports that she wept while addressing the court, but said Knox was upset as <t> she <t> recounted " the pressure, the aggressiveness of the police who called her a liar . "
(a) she is the other family member of lawyer
(b) she is a lawyer
(c) she has no known relations to lawyer
**Analysis**: In the sentence, the word 'she' refers to someone who was upset while recounting certain events in court. The word 'lawyer' refers to someone who denied a news report about that same person weeping in court. There is no information in the sentence to indicate that the two individuals are related in any way. So she has no known relations to lawyer.
**Answer**: (c)

**Instruct**: Read the sentence and determine the relation between MEF and Myanmar Equestrian Federation quoted by <t>.
**Sentence**: YANGON , Dec. 27 -LRB- Xinhua -RRB- – Myanmar will hold a horse race in Yangon to commemorate the country 's 63rd Anniversary Independence Day , the <t> Myanmar Equestrian Federation <t> -LRB- <t> MEF <t> -RRB- confirmed to Xinhua on Monday.
(a) MEF is also known as Myanmar Equestrian Federation
(b) MEF has political affiliation with Myanmar Equestrian Federation
(c) MEF has no known relations to Myanmar Equestrian Federation
**Analysis**: The symbols -LRB- and -RRB- in the sentence stand for left and right round brackets and are used to enclose the abbreviation 'MEF' to indicate that it is a replacement for the longer name 'Myanmar Equestrian Federation. So MEF is also known as Myanmar Equestrian Federation.
**Answer**: (a)

**Instruct**: Read the sentence and determine the relation between Douglas Flint and chairman quoted by <t>.
**Sentence**: At the same time , Chief Financial Officer <t> Douglas Flint <t> will become <t> chairman <t> , succeeding Stephen Green who is leaving to take a government job.
(a) Douglas Flint has no known relations to chairman
(b) Douglas Flint is a chairman
(c) Douglas Flint is the employee of chairman
**Analysis**: The sentence states that Chief Financial Officer Douglas Flint Douglas Flint will succeed Stephen Green as a chairman. So Douglas Flint is a chairman.
**Answer**: (b)

**Instruct**: Read the sentence and determine the relation between FAA and U.S. quoted by <t>.
**Sentence**: On its Web site , the <t> U.S. <t> <t> FAA <t> says the Category 2 rating means the country lacks the laws or regulations that are needed for the certification and oversight of air carriers , according to minimum international standards.
(a) FAA is also known as U.S.
(b) FAA has no known relations to U.S.
(c) FAA has a headquarter in the country U.S.
**Analysis**: The sentence states that the FAA says the Category 2 rating means the country lacks the laws or regulations needed for the certification and oversight of air carriers, indicating that the FAA is responsible for overseeing aviation regulations in the country. Actually the FAA (Federal Aviation Administration) is a U.S. government agency responsible for regulating and overseeing civil aviation in the United States, and it has its headquarters in Washington, D.C.. So FAA has a headquarter in the country U.S..
**Answer**: (c)

Table 18: Demo examples used in ACE05 dataset. We color the trigger word in blue.

**Instruct**: Read following sentences and identify what event is triggered by the word loan quoted by <t>.
**Sentence**: Separately , former WorldCom CEO Bernard Ebbers failed on April 29 to make a first repayment of 25 million dollars, plus interest, on a 400-million - dollar <t> loan <t> from MCI, the Journal said, citing SEC documents.
(a) The word loan does not trigger any known event.
(b) The word loan triggers a TRANSFER-MONEY event: giving, receiving, borrowing, or lending money when it is NOT in the context of purchasing something.
(c) The word loan triggers a DECLARE-BANKRUPTCY event: an Entity officially requests legal protection from debt collection due to an extremely negative balance sheet.
**Analysis**: In the given sentence, the word loan is used to describe the 400-million-dollar loan from MCI to former WorldCom CEO Bernard Ebbers, which he failed to repay on time. This situation clearly involves a transfer of money without the context of purchasing something, which falls under the TRANSFER-MONEY event. So the word loan triggers a TRANSFER-MONEY event: giving, receiving, borrowing, or lending money when it is NOT in the context of purchasing something.
**Answer**: (b)

**Instruct**: Read following sentences and identify what event is triggered by the words treated quoted by <t>.
**Sentence**: When she 's in Germany , Lynch will be <t> treated <t> for bullet wounds and broken bones .
(a) The word treated triggers an INJURE event: a PERSON gets/got injured whether it occurs accidentally, intentionally or even self-inflicted.
(b) The word treated does not trigger any known event.
(c) The word treated triggers a TRANSPORT event: an ARTIFACT (WEAPON or VEHICLE) or a PERSON is moved from one PLACE (GEOPOLITICAL ENTITY, FACILITY, LOCATION) to another.
**Analysis**: The sentence suggests that Lynch has already been injured and will receive medical treatment in Germany for her injuries. The word 'treated' simply describes the medical care she will receive and does not indicate a new event or action taking place. So the word treated does not trigger any known event.
**Answer**: (b)

**Instruct**: Read following sentences and identify what event is triggered by the words buy quoted by <t>.
**Sentence**: And I won't dwell on the irony of an Oracle employee being driven out of Oracle , starting his own company , and forcing Ellison to spend $ 10.3 billion to get his company – but not him – back ( though it does rather delightfully remind me of Coca - Cola basically giving away the bottling franchise and then spending billions to <t> buy <t> it back ) .
(a) The word buy triggers a DECLARE-BANKRUPTCY event: an Entity officially requests legal protection from debt collection due to an extremely negative balance sheet.
(b) The word buy triggers a TRANSFER-OWNERSHIP event: The buying, selling, loaning, borrowing, giving, or receiving of artifacts or organizations by an individual or organization.
(c) The word buy does not trigger any known event.
**Analysis**: In the given sentence, the word buy is used to describe the action of Oracle spending $10.3 billion to get a company back. This clearly involves the transfer of ownership of the company from one entity to another. So the word buy triggers a TRANSFER-OWNERSHIP event: The buying, selling, loaning, borrowing, giving, or receiving of artifacts or organizations by an individual or organization.
**Answer**: (b)

**Instruct**: Read following sentences and identify what event is triggered by the words set quoted by <t>.
**Sentence**: British forces also began establishing the country's first postwar administration Tuesday, granting a local sheik power to <t> set <t> up an administrative committee representing the groups in the region.
(a) The word set triggers a START-POSITION event: a PERSON elected or appointed begins working for (or changes offices within) an ORGANIZATION or GOVERNMENT.
(b) The word set triggers a START-ORG event: a new ORGANIZATION is created.
(c) The word set does not trigger any known event.
**Analysis**: The phrase 'set up' specifically implies the creation or establishment of a new organization or entity, rather than simply the word 'set'. So the word set does not trigger any known event.
**Answer**: (c)

Table 19: Templates for FewNERD dataset, where {ent} is the placeholder for entity type.

| Entity | Template |
|---|---|
| no-entity | {ent} do/does not belong to any known entities. |
| person-artist/author | {ent} is an artist or author. |
| person-actor | {ent} is an actor. |
| art-writtenart | {ent} is a kind of writtenart. |
| person-director | {ent} is a director. |
| person-other | {ent} is a person, but not affiliated with following professions: actor, artist, athlete, author, director, politician, scholar, soldier. |
| organization-other | {ent} pertains to an organization that does not fall under the categories of company, educational institution, government, media, political party, religion, sports league, sports team, band or musical group. |
| organization-company | {ent} is a company |
| organization-sportsteam | {ent} is a sports team |
| organization-sportsleague | {ent} is a sports league |
| product-car | {ent} is a kind of car |
| event-protest | {ent} refers to a protest, uprising or revolution event |
| organization-government/governmentagency | {ent} refers to a government or governmental agency |
| other-biologything | {ent} is a special term about biology / life science. |
| location-GPE | {ent} is a kind of geopolitical entity |
| location-other | {ent} is a geographic locaton that does not fall under the categories of geopolitical entity, body of water, island, mountain, park, road, railway and transit. |
| person-athlete | {ent} is an athlete or coach. |
| art-broadcastprogram | {ent} is a broadcast program. |
| product-other | {ent} is a kind of product that does not fall under the categories of airplane, train, ship, car, weapon, food, electronic game and software. |
| building-other | {ent} is a kind of building that does not fall under the categories of airport, hospital, hotel, library, restaurant, sports facility and theater |
| product-weapon | {ent} is a kind of weapon. |
| building-airport | {ent} is an airport. |
| building-sportsfacility | {ent} is a sports facility building. |
| person-scholar | {ent} is a scholar. |
| art-music | {ent} is a music. |
| event-other | {ent} refers to some event except attack, election, natural disaster, protest, revolution and sports |
| other-language | {ent} is a kind of human language. |
| other-chemicalthing | {ent} is some special term about chemical science. |
| art-film | {ent} is a film. |
| building-hospital | {ent} is a hospital. |
| other-law | {ent} is a legal document, a term or a convention in legal sense. |
| product-airplane | {ent} is kind of airplane product. |
| location-road/railway/highway/transit | {ent} is a geographic position about roadways, railways, highways or public transit systems. |
| person-soldier | {ent} is a soldier |
| location-mountain | {ent} is geographic position about mountain. |
| organization-education | {ent} is an educational institute/organization. |
| organization-media/newspaper | {ent} is a media/newspaper organization. |

| | |
|---|---|
| product-software | {ent} is a software product. |
| location-island | {ent} is geographic position about island. |
| location-bodiesofwater building-library | {ent} is geographic position situated near a body of water. {ent} is a library. |
| other-astronomything | {ent} is a special term about astronomy. |
| person-politician | {ent} is a politician or lawyer or judge. |
| building-hotel | {ent} is a hotel building. |
| product-game | {ent} is a electronic game product. |
| other-award | {ent} is a kind of award. |
| event-sportsevent | {ent} refers to some event related to sports. |
| organization-showorganization | {ent} is a band or musical organization. |
| other-educationaldegree | {ent} is a kind of educational degree. |
| building-theater | {ent} is a theater. |
| other-disease | {ent} is a kind of disease. |
| event-election | {ent} is an event about election. |
| organization-politicalparty | {ent} is a political party/organization. |
| other-currency | {ent} is a kind of currency. |
| event-attack/battle/war/militaryconflict | {ent} is an event about attack, battle, war or military conflict. |
| product-ship | {ent} is a ship. |
| building-restaurant | {ent} is a restaurant. |
| other-livingthing | {ent} is a living animal/creature/organism. |
| art-other | {ent} is a work of art, but not belong to the categories of music, film, written art, broadcast or painting. |
| event-disaster | {ent} is a natural disaster event. |
| organization-religion | {ent} is a religious organization. |
| other-medical | {ent} refers to some kind of medicine.entity |
| location-park | {ent} is a park. |
| other-god | {ent} is a god in some legend/religious story. |
| product-food | {ent} is a kind of food. |
| product-train | {ent} is a kind of train(vehicle). |
| art-painting | {ent} is an art painting. |

Table 20: Templates for TACREV dataset, where {subj} and {obj} are the placeholders for subject and object entities. Copied from (Lu et al., 2022a)

| Relation | Template |
|---|---|
| no_relation | {subj} has no known relations to {obj} |
| per:stateorprovince_of_death | {subj} died in the state or province {obj} |
| per:title | {subj} is a {obj} |
| org:member_of | {subj} is the member of {obj} |
| per:other_family | {subj} is the other family member of {obj} |
| org:country_of_headquarters | {subj} has a headquarter in the country {obj} |
| org:parents | {subj} has the parent company {obj} |
| per:stateorprovince_of_birth | {subj} was born in the state or province {obj} |
| per:spouse | {subj} is the spouse of {obj} |
| per:origin | {subj} has the nationality {obj} |
| per:date_of_birth | {subj} has birthday on {obj} |
| per:schools_attended | {subj} studied in {obj} |
| org:members | {subj} has the member {obj} |
| org:founded | {subj} was founded in {obj} |
| per:stateorprovinces_of_residence | {subj} lives in the state or province {obj} |
| per:date_of_death | {subj} died in the date {obj} |
| org:shareholders | {subj} has shares hold in {obj} |
| org:website | {subj} has the website {obj} |
| org:subsidiaries | {subj} owns {obj} |
| per:charges | {subj} is convicted of {obj} |
| org:dissolved | {subj} dissolved in {obj} |
| org:stateorprovince_of_headquarters | {subj} has a headquarter in the state or province {obj} |
| per:country_of_birth | {subj} was born in the country {obj} |
| per:siblings | {subj} is the siblings of {obj} |
| org:top_members/employees | {subj} has the high level member {obj} |
| per:cause_of_death | {subj} died because of {obj} |
| per:alternate_names | {subj} has the alternate name {obj} |
| org:number_of_employees/members | {subj} has the number of employees {obj} |
| per:cities_of_residence | {subj} lives in the city {obj} |
| org:city_of_headquarters | {subj} has a headquarter in the city {obj} |
| per:children | {subj} is the parent of {obj} |
| per:employee_of | {subj} is the employee of {obj} |
| org:political/religious_affiliation | {subj} has political affiliation with {obj} |
| per:parents | {subj} has the parent {obj} |
| per:city_of_birth | {subj} was born in the city {obj} |
| per:age | {subj} has the age {obj} |
| per:countries_of_residence | {subj} lives in the country {obj} |
| org:alternate_names | {subj} is also known as {obj} |
| per:religion | {subj} has the religion {obj} |
| per:city_of_death | {subj} died in the city {obj} |
| per:country_of_death | {subj} died in the country {obj} |
| org:founded_by | {subj} was founded by {obj} |

Table 21: Templates for ACE05 dataset, where {evt} is the placeholder for event type.

| Event | Template |
|---|---|
| no-event | The word {evt} does not trigger any known event. |
| Movement.Transport | The word {evt} triggers a TRANSPORT event: an ARTIFACT (WEAPON or VEHICLE) or a PERSON is moved from one PLACE (GEOPOLITICAL ENTITY, FACILITY, LOCATION) to another. |
| Personnel.Elect | The word {evt} triggers an ELECT event which implies an election. |
| Personnel.Start-Position | The word {evt} triggers a START-POSITION event: a PERSON elected or appointed begins working for (or changes offices within) an ORGANIZATION or GOVERN-MENT. |
| Personnel.Nominate | The word {evt} triggers a NOMINATE event: a PERSON is proposed for a position through official channels. |
| Conflict.Attack | The word {evt} triggers an ATTACK event: a violent physical act causing harm or damage. |
| Personnel.End-Position | The word {evt} triggers an END-POSITION event: a PERSON stops working for (or changes offices within) an ORGANIZATION or GOVERNMENT. |
| Contact.Meet | The word {evt} triggers a MEET event: two or more entities come together at a single location and interact with one another face-to-face. |
| Life.Marry | The word {evt} triggers a MARRY event: two people are married under the legal definition. |
| Contact.Phone-Write | The word {evt} triggers a PHONE-WRITE event: two or more people directly engage in discussion which does not take place 'face-to-face'. |
| Transaction.Transfer-Money | The word {evt} triggers a TRANSFER-MONEY event: giving, receiving, borrowing, or lending money when it is NOT in the context of purchasing something. |
| Justice.Sue | The word {evt} triggers a SUE event: a court proceeding has been initiated for the purposes of determining the liability of a PERSON, ORGANIZATION or GEOPO-LITICAL ENTITY accused of committing a crime or neglecting a commitment |
| Conflict.Demonstrate | The word {evt} triggers a DEMONSTRATE event: a large number of people come together in a public area to protest or demand some sort of official action. For eample: protests, sit-ins, strikes and riots. |
| Business.End-Org | The word {evt} triggers an END-ORG event: an ORGANIZATION ceases to exist (in other words, goes out of business). |
| Life.Injure | The word {evt} triggers an INJURE event: a PERSON gets/got injured whether it occurs accidentally, intentionally or even self-inflicted. |
| Life.Die | The word {evt} triggers a DIE event: a PERSON dies/died whether it occurs acci-dentally, intentionally or even self-inflicted. |
| Justice.Arrest-Jail | The word {evt} triggers a ARREST-JAIL event: a PERSON is sent to prison. |
| Transaction.Transfer-Ownership | The word {evt} triggers a TRANSFER-OWNERSHIP event: The buying, selling, loaning, borrowing, giving, or receiving of artifacts or organizations by an individual or organization. |
| Justice.Execute | The word {evt} triggers an EXECUTE event: a PERSON is/was executed |
| Justice.Trial-Hearing | The word {evt} triggers a TRIAL-HEARING event: a court proceeding has been initiated for the purposes of determining the guilt or innocence of a PERSON, ORGANIZATION or GEOPOLITICAL ENTITY accused of committing a crime. |
| Justice.Sentence | The word {evt} triggers a SENTENCE event: the punishment for the DEFENDANT is issued |
| Life.Be-Born | The word {evt} triggers a BE-BORN event: a PERSON is given birth to. |
| Justice.Charge-Indict | The word {evt} triggers a CHARGE-INDICT event: a PERSON, ORGANIZATION or GEOPOLITICAL ENTITY is accused of a crime |
| Business.Start-Org | The word {evt} triggers a START-ORG event: a new ORGANIZATION is created. |
| Justice.Convict | The word {evt} trigges a CONVICT event: a PERSON, ORGANIZATION or GEOPOLITICAL ENTITY is convicted whenever it has been found guilty of a CRIME. |
| Business.Declare-Bankruptcy | The word {evt} triggers a DECLARE-BANKRUPTCY event: an Entity officially requests legal protection from debt collection due to an extremely negative balance sheet. |
| Justice.Release-Parole | The word {evt} triggers a RELEASE-PAROLE event. |

| | |
|---|---|
| Justice.Fine | The word {evt} triggers a FINE event: a GEOPOLITICAL ENTITY, PERSON or ORGANIZATION get financial punishment typically as a result of court proceedings. |
| Justice.Pardon | The word {evt} triggers a PARDON event: a head-of-state or their appointed representative lifts a sentence imposed by the judiciary. |
| Justice.Appeal | The word {evt} triggers a APPEAL event: the decision of a court is taken to a higher court for review |
| Business.Merge-Org | The word {evt} triggers a MERGE-ORG event: two or more ORGANIZATION Entities come together to form a new ORGANIZATION Entity. |
| Justice.Extradite | The word {evt} triggers a EXTRADITE event. |
| Life.Divorce | The word {evt} triggers a DIVORCE event: two people are officially divorced under the legal definition of divorce. |
| Justice.Acquit | The word {evt} triggers a ACQUIT event: a trial ends but fails to produce a conviction. |