# OpenReview forum: "Large Language Model Is Not a Good Few-shot Information Extractor, but a Good Reranker for Hard Samples!"
_EMNLP/2023/Conference — EMNLP 2023 Findings_

### Official Review · Reviewer_CUTz · 2023-08-03

**Typos Grammar Style And Presentation Improvements:** none
**Soundness:** 3

**Excitement:**

3: Ambivalent: It has merits (e.g., it reports state-of-the-art results, the idea is nice), but there are key weaknesses (e.g., it describes incremental work), and it can significantly benefit from another round of revision. However, I won't object to accepting it if my co-reviewers champion it.

**Missing References:**

none

**Paper Topic And Main Contributions:**

This paper investigates the use of Large Language Models (LLMs) as few-shot solvers for information extraction (IE) tasks and their comparison with fine-tuned small Pre-trained Language Models (SLMs).  The main contributions include extensive experiments on multiple datasets, highlighting that LLMs are not effective few-shot information extractors but can complement SLMs by handling difficult samples.  The paper proposes an adaptive filter-then-rerank paradigm where SLMs filter challenging samples, and LLMs rerank them to achieve promising improvements (average 2.4% F1-gain) on various IE tasks while maintaining reasonable time and cost efficiency.  The research contributes insights into building efficient and effective IE systems by leveraging both LLMs and SLMs.

**Reasons To Accept:**

1. The paper provides a thorough evaluation of LLMs as few-shot solvers for IE tasks, comparing them to fine-tuned SLMs on multiple datasets and across various IE tasks.  This extensive evaluation helps the NLP community gain a deeper understanding of the capabilities and limitations of LLMs in this context.

2. The proposed adaptive filter-then-rerank paradigm, leveraging the complementary strengths of LLMs and SLMs, offers a novel and practical solution to improve IE performance while managing time and budget constraints.

**Reasons To Reject:**

The paper revolves around InstructGPT and GPT-4 to discuss the filter-then-rerank paradigm, and there are two potential problems:
1. It is not certain that the training set was used in the pre-training process of the two LLMs, and this matter is not discussed in the paper.

2. The validity of the filter-then-rerank paradigm is not discussed in the context of small-scale generative language models such as LLaMA and Vicuna in Figure 2.

In addition, some of the conclusions in the paper (such as those corresponding to lines 64 to 70) are not exciting, or even almost taken for granted.

**Reproducibility:**

5: Could easily reproduce the results.

**Reviewer Confidence:**

4: Quite sure. I tried to check the important points carefully. It's unlikely, though conceivable, that I missed something that should affect my ratings.

---

> ### Author Rebuttal · Authors · 2023-08-28
>
> Thanks a lot for your reviews! Your professional reviews offer us great advice towards writing a more comprehensive and competitive paper! And, we appreciate you finding our empirical study extensive and our proposed filter-then-rerank paradigm combining the strength of both SLMs and LLMs.
>
> ---
>
> > Q1: It is not certain that the training set was used in the pre-training process of the two LLMs, and this matter is not discussed in the paper.
>
> Reducing the effect of data contamination when evaluating LLMs is an open, challenging and not well-solved problem. It is beyond the main scope of this paper. However, **we still discuss this potential problem in the Limitations part (lines 519 to 528)**. We argue that this potential problem does **NOT** affect the validity of the filter-then-rerank paradigm because:
> 1. Data contamination makes the results overestimated. And the inferior overall performance of LLMs on our experiments (see Figure 2) demonstrates that the extent of data contamination on these datasets is limited.
> 2. A reasonable assumption is that the effect of data contamination is equal on hard samples and easy samples. Since we observe a distinct performance difference on both easy and hard samples, we believe LLMs’ proficiencies on hard samples are due to their parametric knowledge and reasoning abilities (Line 85-88) instead of sample  leakage (otherwise LLMs should perform well on easy samples as well).
>
> ---
> ---
>
> > Q2: The validity of the filter-then-rerank paradigm is not discussed in the context of small-scale generative language models such as LLaMA and Vicuna in Figure 2.
>
> We appreciate your valuable suggestion to discuss the validity of the filter-then-rerank paradigm on smaller, open-source LLMs. We add experiments with Vicuna-13B, which is instruction-tuned on LLaMA-13B, as the reranker and extend the Table 3 as below (We do not run experiments on LLaMA because we require the rerankers being LLMs with instruction tuning, on which the multi-choice question (MCQ) prompts could leverage their powers). We conduct 3-run experiments and report their mean values and standard deviations (in bracket).
>
> | | NER-FewNERD (5-shot) | NER-FewNERD (10-shot) |  NER-FewNERD (20-shot) | RE-TACREV (20-shot) | RE-TACREV (50-shot) | RE-TACREV (100-shot) | ED - ACE (5-shot) | ED - ACE (10-shot) | ED - ACE (20-shot) |
> | --- | --- | --- | --- | --- | --- | --- | --- | --- | --- |
> | SLM | 59.4 (1.5) | 61.4 (0.8) | 61.9 (1.2) | 62.4 (3.8) | 68.5 (1.6) | 72.6 (1.5) | 55.1 (4.6) | 63.9 (0.8) | 65.8 (2.0) |
> | SLM + Vicuna| 60.0 (1.8) | 61.9 (2.1) | 62.2 (1.4) | 65.2 (1.4) | 70.8 (1.6) | 73.8 (1.7) | 56.9 (4.0) | 63.5 (2.7) | 66.0 (2.6) |
> | SLM w. ensemble | 59.6 (1.7) | 61.8 (1.2) | 62.6 (1.0) | 64.9 (1.5) | 71.9 (2.2) | 74.1 (1.7) | 56.9 (4.7) | 64.2 (2.1) | 66.5 (1.7) |
> | SLM w. ensemble + Vicuna | 59.9 (0.7) | 62.1 (0.7) |62.8 (1.1) |66.5 (0.5) | 73.6 (1.4) |75.0(1.5) | 57.9 (5.2) | 64.4 (1.2) |66.2 (2.4) |
>
> We observe that:
> 1. **Our filter-then-rerank paradigm still works and outperforms SLM-only approaches under most cases.**
> 2. The reranking gains from Vicuna-13B are significantly less than that of GPT-4 (0.8% v.s. 2.4% absolute F1-gain, on average).
> 3. Vicuna-13B presents inconsistent reranking capabilities among different tasks. It shows strongest performance on RE task, while shows minimal even negative effects on ED task.
>
> We argue such observation does **NOT** undermine the validity of our filter-then-rerank paradigm because
> 1. Being a LLaMA-based model distilled from GPT-4 auto-generated data, Vicuna has its limitations on both the pre-training and instruction-tuning scale. **It is reasonable that Vicuna presents inferior and less balanced performance than GPT-4.** We believe more powerful open-source LLMs would be likely to achieve stronger performance than Vicuna.
> 2. As stated in Section 5.4, we only call LLMs for a small portion of samples. So the overall time and budget cost is affordable even with GPT-4 as the reranker.
>
> ---
> ---
>
> > Q3: Some of the conclusions in the paper (such as those corresponding to lines 64 to 70) are not exciting, or even almost taken for granted.
>
> 1. The conclusions shown in lines 64 to 70 are somewhat unsurprising from the posterior perspective (after reading our paper). Even if some guesses seem reasonable and intuitive, however, they still require empirical studies to support or falsify them to be a conclusion. To our best knowledge, this paper is the first work to conduct extensive and systematic experiments to evaluate the performance of LLMs on diverse IE tasks. **It provides supportive references for the imprecise intuitions of the community and is valuable from this point**.
>
> 2. Additionally, conclusions in lines 64 to 70 are only some of conclusions in our work and by no means encompass the full breadth of conclusions drawn from our empirical analyses. For example, (1) we explore the effect of various demonstration selection strategies, demonstration number and prompt formats in Section 3.3, (2) we observe that LLMs are relatively more proficient on hard samples than easy samples in Section 4.2, and (3) a detailed analysis and case study on our filter-then-rerank paradigm in Section 5.4 and 5.6.
>
> ---
>
> We hope that our explanations and newly-added experiments have addressed your concern about (1) the potential data contamintation problem, (2) the validity of filter-then-rerank paradigm on smaller, open-source LLMs, and (3) the novelty and excitement of our work. If not, would you mind telling us your remaining concerns? We would try our best efforts to further address your any concerns.

---

### Official Review · Reviewer_ifw3 · 2023-08-03

**Soundness:** 4

**Excitement:**

3: Ambivalent: It has merits (e.g., it reports state-of-the-art results, the idea is nice), but there are key weaknesses (e.g., it describes incremental work), and it can significantly benefit from another round of revision. However, I won't object to accepting it if my co-reviewers champion it.

**Paper Topic And Main Contributions:**

In this thorough study, the authors set out to determine whether LLMs can solve information extraction tasks effectively or whether these tasks are still dominated by smaller LMs. In a broad and systematic evaluation over 9 datasets and 4 IE tasks, they provide evidence that training small models through different techniques is still the most effective way to perform well on IE tasks. Especially since strong performance is achieved with fewer than 100 examples and small models (RoBERTa-large and T5-large).

Apart from a solid empirical study, the paper explores the idea that SLMs are better than LLMs at easier examples while LLMs are better at more complex examples (if the conditions of the example are simplified first by the smaller supervised model). The authors then propose a new method which leverages this insight by using the SLM for easy samples and LLMs for hard samples. This approach yields large improvements in examples which are challenging for SLMs while still leveraging the strengths of trained models in high-confidence examples.

**Questions For The Authors:**

- Could you add detailed statistics around your test subsets? I know you downsampled randomly but I would like to know the relation percentage as well as the negative vs positive ratio in the different datasets (previous work has found that this is very important for IE performance).

**Reasons To Accept:**

- Systematic, comprehensive and timely empirical study
- Provides some insight into what IE examples and tasks are challenging for LLMs
- Proposes a way to utilize each model type optimally by keeping performance up and costs down (strong practical contribution)

**Reasons To Reject:**

- The empirical and practical contributions of this paper are robust and I would like it to be accepted to the conference.
- Nevertheless, this paper’s breadth does somewhat hurt its depth. It is unfortunate that the authors were unable to do a more thorough analysis on why LLMs are failing on easy samples. A case study on this or at least some deeper discussion on this topic could greatly improve the paper.

**Reproducibility:**

4: Could mostly reproduce the results, but there may be some variation because of sample variance or minor variations in their interpretation of the protocol or method.

**Reviewer Confidence:**

4: Quite sure. I tried to check the important points carefully. It's unlikely, though conceivable, that I missed something that should affect my ratings.

**Typos Grammar Style And Presentation Improvements:**

- “Strength (and deficiency)” L076
- “Instruction I: The instruction locates at the very” L219
- “(Identity the entities...in the sentence)” L221
- For in-text section titles like “LLMs show limited inference speed” in L275, place a period after it unless the title is above. Looks a bit messy otherwise.
- L282: “Discussion: Why do LLMs not Present Satisfactory Performance on IE Tasks?” -> “Discussion: Why LLMs Fail to Obtain Satisfactory Performance on IE Tasks?” (Title sounds a bit awkward)
- “labels, then LLMs subsequently rerank.” L325
- “LLMs are and only are Hard Sample Solver” L343 (sounds awkward)
- “with low confidence scores as hard otherwise easy.” L364

---

> ### Author Rebuttal · Authors · 2023-08-28
>
> Thanks a lot for your reviews! Your professional reviews offer us great advice towards writing a more comprehensive and competitive paper! And, we appreciate you finding our empirical study covering broadly and our proposed filter-then-rerank paradigm practical.
>
> -----------------
>
> > Q1: The authors did not do a more thorough analysis on why LLMs are failing on easy samples. A case study on this or at least some deeper discussion on this topic could greatly improve the paper.
>
> Thanks for your constructive suggestion. We summarize the reason for LLMs' failure on easy samples from two aspects: (1) SLMs’ remarkable and better performance on easy samples. (2) LLMs’ predisposition to false-positive predictions on negative samples (most of which are easy samples). The tendency towards false-positive predictions, specifically, are due to (a) LLM hallucination and (b) Span-boundary mismatch. We provide more detailed analysis and case studies as below:
>
> 1. The performance of LLMs (red lines in Figure 4, the same below) are more stable than that of SLMs (blue lines) among samples with various difficulties (LLMs’ curves are more flatten). Specifically, SLMs trail LLMs on hard samples (left parts of the figures) but gradually surpass LLMs on easier samples (right parts of the figures). Therefore, the LLMs’ failure on easy samples are largely **relative**. In other words, we could find a better alternative, i.e. SLM-based methods, to solve the hard samples.
> 2. On NER and ED tasks, however, we indeed observe **absolute** performance drops when the samples are extremely easy (see the blue curves’ drops in the first and third subfigures at the very right parts in Figure 4). Since NER and ED are sequence-labeling tasks, i.e. determining the label type of every word/span in the sentence, almost all samples (words/spans) are actually negative samples. Also, **most of these negative samples are extremely easy for SLMs** (see the last table in Q2, in which the sample-level negative-positive ratios in easy samples are much higher than hard samples). Due to LLMs’ **predisposition to false-positive predictions on negative samples**, the performance of LLMs on easy samples collapses. Specifically, we attribute such false-positive predictions to two main reasons: (1) LLM hallucination and (2) span boundary mismatch. We take NER task as an example and detail such two kinds of mistakes with cases as below:
>
>     2(a). **LLM Hallucination**: the most distinctive mistakes made by LLMs. Without specific fine-tuning on IE tasks, LLMs (with in-context learning approaches) sometimes hallucinate and provide meaningless, even confusing explanations or answers. For example, LLMs might predict a verb, preposition or even special symbols as an entity. Here are two cases:
>
>     Case (1): LLMs predict the word ```die``` as entity ```person-other``` (False-positive prediction, ground truth should be ```None```)
>     ```
>     Sentence: Flavius Rumoridus ( died 5th century AD ) was a Roman soldier who was appointed consul in AD 403 in the Western Roman Empire at the same time as the eastern emperor Theodosius II served in the same capacity in the East .
>     Candidate Choices:
>     (a): died do/does not belong to any known entities.
>     (b): died is a kind of geopolitical entity
>     (c): died is a person, but not affiliated with following professions: actor, artist, athlete, author, director, politician, scholar, soldier.
>
>     Analysis: The sentence provides information about Flavius Rumoridus, who was a Roman soldier and appointed consul in AD 403. The sentence also mentions that he died in the 5th century AD, which is a statement about his death. Therefore, the correct answer is (c) because the sentence does not provide any information about Flavius Rumoridus being affiliated with any of the professions listed in option (c).
>     Correct Answer: (c)
>     ```
>
>     Case (2): LLMs predict the word ```of``` as entity ```organization-other``` (False-positive prediction, ground truth should be ```None```)
>     ```
>     Sentence: During these years he studied at the South Kensington Schools of the Royal College of Art .
>     Candidate Choices:
>     (a): of do/does not belong to any known entities.
>     (b): of is an educational institute/organization.
>     (c): of pertains to an organization that does not fall under the categories of company, educational institution, government, media, political party, religion, sports league, sports team, band or musical group.
>
>     Analysis: In the given sentence, \"of\" is a preposition that indicates the relationship between \"South Kensington Schools\" and \"Royal College of Art.\" None of the candidate choices directly address the correct relationship between these entities or state that they are not applicable. Therefore, the correct answer is (c): \"of\" does not pertain to any known entities mentioned in the choices provided.
>     Correct Answer: (c)
>     ```
>
>     2(b). **Span boundary mismatch**: Different from sentence-level approaches which take one or multiple sentences as an unit and assign each word’s label globally, LLMs under our filter-then-rerank paradigm adopt sample-level approaches (lines 326-330). In other words, LLMs only consider each word/span candidate independently. The sample-level approaches enable LLMs to rerank candidates with the help of multi-choice question prompt. However, they also aggravate the span boundary mismatching problem in NER and ED tasks. We empirically find easy samples suffering much more from boundary mismatching problem than hard samples. Here are two cases:
>
>     Case (3): LLMs predict  ```Baron```, a subspan of ```Baron Carl Hurleman```, as the entity ```person-other```. It is a false-positive prediction, the ground-truth entity type of ```Baron``` is ```None```. And the ground-truth entity type of ```Baron Carl Hurleman``` is ```person-artist/author```.
>     ```
>     Sentence: Additions in the 1740s were by Baron Carl Hurleman ( 1700 - 1753 ) .
>     Candidate Choices:
>     (a): Baron do/does not belong to any known entities.
>     (b): Baron is a person, but not affiliated with following professions: actor, artist, athlete, author, director, politician, scholar, soldier.
>     (c): Baron is a kind of geopolitical entity
>
>     Analysis: In the given sentence, \"Baron\" is a title given to a person named Carl Hurleman, who is mentioned as the one who made additions in the 1740s. The sentence does not provide any information about connections to geopolitical entities or any other known entities mentioned in the choices provided. Therefore, the correct answer is (b): Baron is a person, but not affiliated with following professions: actor, artist, athlete, author, director, politician, scholar, soldier.
>     Correct Answer: (b)
>     ```
>
>     Case (4): LLMs predict ```65th```, a subspan of ```65th British Academy Film Awards```, as the entity ```other-awards```. It is a false-positive prediction, the ground-truth entity type of ```65th``` is ```None```. And the ground-truth entity type of ```65th British Academy Film Awards``` is ```other-awards```.
>
>     ```
>     Sentence: Ralph Fiennes was nominated for the BAFTA Award for Outstanding Debut by a British Writer , Director or Producer at the 65th British Academy Film Awards .
>     Candidate Choices:
>     (a): 65th do/does not belong to any known entities.
>     (b): 65th is a kind of award.
>     (c): 65th refers to some event except attack, election, natural disaster, protest, revolution and sports
>
>     Analysis: The sentence mentions the \"65th British Academy Film Awards,\" which is an event that recognizes outstanding achievements in the film industry. Therefore, option (c) is incorrect as it excludes events like award ceremonies. Option (a) is also incorrect as the number \"65th\" is associated with the event. Hence, the correct answer is (b) as \"65th\" refers to the specific edition of the BAFTA Awards.
>     Correct Answer: (b)
>     ```
>
> Based on the analysis above, we would add an extra subsection in Section 4 to discuss why LLMs work on hard samples but fail on easy samples. Specifically, we would combine the current Section 5.6 (LLMs on hard samples, with cases in Appendix E.1) and our response for Q1 (LLMs on easy samples). We hope that this new subsection (1) illustrates clearer about the advantages and disadvantages of LLMs and SLMs, and (2) provides a more meticulous and comprehensive motivation explanation of our proposed filter-then-rerank paradigm.
>
> --------------------------
> -----------------
>
> > Q2:  Could you add detailed statistics around your test subsets? I know you downsampled randomly but I would like to know the relation percentage as well as the negative vs positive ratio in the different datasets (previous work has found that this is very important for IE performance).
>
> We appreciate your insightful suggestion. We provide more detailed statistics about the test subsets as below, including their (1) sentence numbers and (2) sample (entity/relation/trigger word/argument for different IE tasks) numbers as below.
>
> | | NER-CoNLL | NER-OntoNotes | NER-FewNERD | RE-TACREV | RE-TACRED | ED-ACE | ED-ERE | ED-MAVEN | EAE-ACE | EAE-ERE | EAE-RAMS |
> | --- | --- | --- | --- | --- | --- | --- | --- | --- | --- | --- | --- |
> | # Sentence| 250 | 250 | 250 | 500 | 500 | 250 | 250 | 250 | 250 | 250 | 250 |
> | # Sample | 388 | 342 | 745 | 376 | 361 | 151 | 176 | 689 | 395 | 731 | 589 |
>
> Regarding negative-positive ratios (NP ratios; we also view the relation percentage in RE task as a special negative-positive ratio), we consider two variants, i.e. (1) sentence-level and (2) sample-level NP ratios. They correspond to the scenarios of our main empirical studies (Section 3) and filter-then-rerank paradigm (Section 4 and 5), respectively. We investigate their effects on LLMs' performance. Briefly speaking, we do observe the remarkable relations of **sample-level** NP ratios with regard to sample difficulty. As mentioned in Q1, the high NP ratios in easy samples and the LLMs' tendency towards false-negative predictions largely cause LLMs failing on easy samples. We detail these two kinds of NP ratios as below:
>
> 1. **Sentence-level**: If a sentence does not include any positive labels, we view this sentence a sentence-level negative sample. Otherwise it is a sentence-level positive sample. We show the sentence-level NP ratios of all datasets we used in our empirical studies in Section 3 as below.
>
>     *When downsampling test subsets, we ensure that each label has at least one corresponding sample to avoid the absence of rare labels in our evaluation.*
>     | | NER-CoNLL | NER-OntoNotes | NER-FewNERD | RE-TACREV | RE-TACRED | ED-ACE | ED-ERE | ED-MAVEN | EAE-ACE | EAE-ERE | EAE-RAMS |
>     | --- | --- | --- | --- | --- | --- | --- | --- | --- | --- | --- | --- |
>     | Sentence-level NP ratio | 0.30 | 1.03 | 0.13 | 3.03 | 2.59 | 1.35 |1.25 | 0.03 | 0.23 | 0.02 | 0.02 |
>
>     We found that there are **no significant correlations between the negative-positive ratio of the datasets and their results on LLM evaluation**. For example, both the CoNLL and MAVEN have few negative sentence-level samples. But LLMs perform much better on CoNLL than on MAVEN (see Figure 2). Compared with other factors such as task difficulty and/or label number, the sentence-level NP ratios are not a determining factor to affect the IE performance on LLMs.
>     However, we admit the importance of the sentence-level NP ratios. To decouple the other confounders among different datasets, we plan to curate multiple test subsets, having different sentence-level NP ratios, for each dataset. We hope that the results on these different test subsets from single original dataset could better reveal the relation of LLMs' IE performance with regard to different sentence-level NP ratios. We promise to have a deeper analysis about the effect of sentence-level NP ratios in our updated version.
>
> 2. **Sample-level**: For NER and ED tasks, each sentence could have zero, one or more entities / trigger words. Therefore LLMs in our filter–then-rerank paradigm adopt another prompt strategy: they predict a single entity/event candidate once a time (lines 326-330).  We call such word/span a sample-level negative sample if it were not any entities or trigger words. Otherwise it were a sample-level positive sample. Given two NER and ED datasets used in Section 4 and 5, we show sample-level NP ratios on them and their easy/hard subsets as below. We set fixed threshold $\tau$ here for simplicity.
>
>     | | NER-FewNERD | ED-ACE |
>     | --- | --- | --- |
>     | Overall | 5.88 | 38.2
>     | (very) Easy samples ($\tau$ > 0.9) | 9.44 | 44.0 |
>     | Hard samples ($\tau$ < 0.6) | 1.28 | 1.36 |
>
>     We observe that **the sample-level NP ratios of easy samples are much higher than that of hard samples**. As we mentioned in Q1, LLMs tend to provide false positive predictions (more likely failing on negative samples). Thus the LLMs under filter-then-reranker paradigm have relatively more competitive performance on samples with low sample-level NP ratio. From this perspective, our classification on hard and easy samples is a simple but effective strategy to lower the NP ratios of samples feeding to LLMs. This strategy greatly avoids the limitations and leverages the powers of LLM rerankers.
>
> -----------------
>
> We additionally appreciate your kind and valuable comments on the typos, grammer, and writing styles of our paper. We would revise our paper to correct these problems. We hope that our explanations and newly-added experiments have addressed your concern about (1) the reasons for LLMs failing on easy samples and (2) the potential effect of negative-positive ratios on LLMs' performance. If not, would you mind telling us your remaining concerns? We would try our best efforts to further address your any concerns.

---

### Official Review · Reviewer_A8RC · 2023-08-10

**Soundness:** 4

**Excitement:**

4: Strong: This paper deepens the understanding of some phenomenon or lowers the barriers to an existing research direction.

**Paper Topic And Main Contributions:**

The authors show that LLMs are not effective few-show information extractors in general, but can well complement SLMs and effectively solve hard samples that SLMs struggle with. They propose an adaptive filter-then-rerank paradigm, which utilizes LLMs to rerank a small portion of difficult samples identified by SLMs.

**Questions For The Authors:**

Please refer to the Reasons To Reject.

**Reasons To Accept:**

1. The paper is well written and easy to understand.
2. The authors thoroughly discussed why LLMs have unsatisfactory performance on information extraction tasks.
3. The proposed filter-then-rerank paradigm is simple but insightful.
4. The experiments are thorough and effectively validate the author's hypotheses. Moreover, experiment results demonstrate hat the proposed method is effective.

**Reasons To Reject:**

1. The instructions used in this paper are simple. Would this directly affect the performance of LLMs on information extraction tasks? Did the authors try other complex instructions?
2. The authors argue that LLMs struggle with fine-grained datasets. Are there any quantitative analyses?

**Reproducibility:**

4: Could mostly reproduce the results, but there may be some variation because of sample variance or minor variations in their interpretation of the protocol or method.

**Reviewer Confidence:**

3: Pretty sure, but there's a chance I missed something. Although I have a good feel for this area in general, I did not carefully check the paper's details, e.g., the math, experimental design, or novelty.

---

> ### Author Rebuttal · Authors · 2023-08-28
>
> Thanks a lot for your reviews! Your professional reviews offer us great advice towards writing a more comprehensive and competitive paper! And, we appreciate you finding our empirical study thorough and our proposed filter-then-rerank paradigm insightful.
>
> -----------------
>
> > Q1: The instructions used in this paper are simple. Would this directly affect the performance of LLMs on information extraction tasks? Did the authors try other complex instructions?
>
> 1. We fully understand your concern that the prompt, especially the instructions, might significantly affect the performance of LLMs on IE tasks. To investigate whether LLMs would benefit from complex instructions, we explored six instruction variants from simple to complex as below. We briefly introduce them as below and show their formats in detail at the bottom of our response.
>
>
>
>     **I0**: No instruction.
>
>     **I1**: Simple instruction, used in our main experiment.
>
>     **I2**: Instruction with label explanation: We concatenate the explanation/definition of each label after their names in the instruction.
>
>     **I3**, **I4**: Instruction with format explanation: We provide two more detailed instructions about the task and the input/output format. These two instructions are written by two authors respectively.
>
>     **I5**: Instruction with both label and format explanation: We combine I2 and I3 to a more complex instruction.
>
>
>     Regarding these six instructions, we evaluate the performance of ChatGPT (gpt-3.5-turbo-0301) on four 20-shot IE tasks. We conduct 3-run experiments and report their mean values and standard deviations (in bracket).
>
>     | | FewNERD (NER)  | TACREV (RE) | ACE (ED) | ACE (EAE) |
>     | --- |  ----  | ----  |  ----  | ----  |
>     | I0 | 57.6 (2.1) | 49.1 (2.4) |  44.0 (1.4) | 50.9 (0.1) |
>     | I1 | 58.3 (0.5)  | 49.6 (1.2) |  42.6 (1.0)  | 51.5 (1.1)  |
>     | I2 | 57.7 (1.0) | 50.0 (1.5) | 41.8 (0.9) | 50.3 (1.5) |
>     | I3 | 57.6 (2.3)  | 52.3 (1.8) |  42.9 (1.3)  | 49.2 (2.3)  |
>     | I4 | 56.8 (0.9)  | 49.6 (2.9) |  41.6 (1.9)  | 49.9 (1.2)  |
>     | I5 | 57.8 (0.5)  | 47.2 (1.7) |  43.1 (1.8)  | 50.6 (1.8)  |
>
>     We observe that there is **no significant correlation between the instruction complexity and the LLMs’ performance**. Even the prompt without instruction (I0) leads to comparable even better performance than prompt with complex instructions.
>
> 2. As we state in lines 196-210, the prompt consists of instruction, demonstrations and the input query. **From our previous experience, the demonstration is the more determined factor than the instruction for the LLMs' performance under few-shot IE task settings.** Therefore, we additionally conduct a series of pivot experiments to find the (nearly) optimal demonstration number, selection strategy and format. We detail them in Appendix D.3, D.4 and D.5 respectively. We adopt the best combination we found in the main experiments.
> 3. In summary, we work hard to find the best prompts, in terms of both their instructions, demonstrations and formats, to evaluate LLMs’ IE abilities. As an empirical study, we acknowledge in the Limitations (lines 509-518) that we do not, and have no possibility, to enumerate all prompt (either instruction or demonstration) combinations. But we believe there unlikely exists a lottery prompt which largely improves the LLMs’ performance and alters our conclusion that LLM performs poorly on few-shot IE tasks.
>
> Instruction 0 (Empty instruction)
> ```
> ```
>
> Instruction 1 (Simple instruction. Take NER task as an example, the same below.)
> ```
> Identify the entities expressed by each sentence, and locate each entity to words in sentence. The possible entity types are:
> [Type_1], [Type_2],.., [Type_N].
> If you do not find any entity in this sentence, just output '''Answer: No entities found.'''
> ```
>
> Instruction 2 (+ label explanation)
> ```
> Identify the entities expressed by each sentence, and locate each entity to words in sentence. The possible entity types are:
> [Type_1]: [Definition_1]
> [Type_2]: [Definition_2]
> ..
> [Type_N]: [Definition_N]
> If you do not find any entity in this sentence, just output '''Answer: No entities found.'''
> ```
>
> Instruction 3 (+ format explanation v1)
> ```
> Assume you are an entity-instance annotator. Given a sentence, you need to (1) identify the word or phrase about the entity in the sentence, and (2) classify its entity type. The possible entity types are listed as below:
> [Type_1], [Type_2], …, [Type_N].
> Please note that your annotation results must follow such format: '''Answer: ([Type_1] <SEP> identified_entity:[Entity_1]), ([Type_2] <SEP> identified_entity:[Entity_2]), ......'''.
> If you do not find any entity in this sentence, just output '''Answer: No entities found.'''
> ```
>
> Instruction 4 (+ format explanation v2)
> ```
> Assume you are an entity-instance annotator. Your objective is to peform a series of intricate steps for Named Entity Recognition. Firstly, you have to identify a particular word or phrase in the sentence that corresponds to an entity. Following this, classify the entity into one of the potential entity types. The potential entity types are provided as below:
> [Type_1], [Type_2], …, [Type_N].
> Please note that your annotation results must follow such format: '''Answer: ([Type_1] <SEP> identified_entity:[Entity_1]), ([Type_2] <SEP> identified_entity:[Entity_2]), ......'''.
> If you do not find any entity in this sentence, just output '''Answer: No entities found.'''
> ```
>
> Instruction 5 (+ label&format explanation)
> ```
> Assume you are an entity-instance annotator. Given a sentence, you need to (1) identify the word or phrase about the entity in the sentence, and (2) classify its entity type. The possible entity types are listed as below:
> [Type_1]: [Definition_1]
> [Type_2]: [Definition_2]
> ..
> [Type_N]: [Definition_N]
> '''
> Please note that your annotation results must follow such format: '''Answer: ([Type_1] <SEP> identified_entity:[Entity_1]), ([Type_2] <SEP> identified_entity:[Entity_2]), ......'''.
> If you do not find any entity in this sentence, just output '''Answer: No entities found.'''
> ```
>
> -----------------
>
> -----------------
>
> > Q2: Are there any quantitative analyses to support that LLMs struggle with fine-grained dataset?
>
> Thanks for your constructive suggestion. Based on the results shown in Figure 2, we additionally provide a quantitative analysis to show that LLMs struggle with fine-grained datasets. Under the 5-shot setting, we compare the performance difference of LLMs (ChatGPT) and SLMs (SoTA few-shot models for each IE task) among different datasets. For each IE task, we observe **a clear negative correlation between the label number (row 2) and the performance difference (row 5)**. In other words, with more label types, **LLMs tend to perform relatively worse than SLMs**. Therefore we conclude that LLMs struggle on fine-grained datasets.
>
> **NER (5-shot, SLM: FSLS, LLM: ChatGPT)**
> |---| CoNLL | OntoNotes | FewNERD |
> |---|---|---|---|
> |#Entity| 4 | 18 | 66 |
> |SLM Performance | 52.5| 59.7 | 59.4 |
> |LLM Performance | 77.8 | 59.4 | 55.5 |
> |$\Delta$ (LLM, SLM)  | 25.3 | -0.3 | -3.9 |
>
>
> **ED (5-shot, SLM: FSLS, LLM: ChatGPT)**
> |---| ACE05 | ERE | MAVEN |
> |---|---|---|---|
> |#Event| 33 | 38 | 168 |
> |SLM Performance | 55.1 | 48.0 | 49.4 |
> |LLM Performance | 39.6 | 33.8 | 25.3 |
> |$\Delta$ (LLM, SLM)  | -15.5 | -14.2 | -24.1 |
>
> **EAE (5-shot, SLM: PAIE, LLM: ChatGPT)**
> |---| ACE05 | ERE | RAMS |
> |---|---|---|---|
> |#Event/#Role| 33/22 | 38/26 | 139/65 |
> |SLM Performance | 45.9 | 40.4 | 54.1 |
> |LLM Performance | 52.8 | 40.7 | 44.2 |
> |$\Delta$ (LLM, SLM)  | 6.9 | 0.3 | -9.9 |
>
> ---
>
> We hope that our explanations and newly-added experiments have addressed your concern about (1) the effect of instructions on LLMs' performance and (2) the quantitative analysis on LLMs' struggling on fine-grained datasets. If not, would you mind telling us your remaining concerns? We would try our best efforts to further address your any concerns.

---

### Meta-Review · Area_Chair_ZMzn · 2023-09-20

**Recommendation:** 4

**Metareview:**

This is a comprehensive empirical study that leads to the observation that LLMs are generally not as competitive for IE tasks than fine-tuned small LMs. The paper then proposes a hybrid strategy that uses LLMs to rerank for "hard" examples that a small LM finds challenging. Empirical results demonstrate the effectiveness of this strategy.

Strengths:
- Solid and comprehensive empirical study on an important problem
- Clear and professional writing
- An intuitively plausible hybrid strategy with demonstrated effectiveness

Weaknesses:
- As the reviewers pointed out, the paper falls short in investigating and explaining why LLMs are underperforming, especially given the known sensitivity to prompts. The authors have responded to this criticism with new analysis and experiments. The authors are strongly encouraged to add these to the revised version.
- As Zhang et al. (2023), which is already cited in this paper, have shown, LLMs' RE performance can be substantially improved by reformulating RE as a QA problem. One possibility demonstrated through that work is, perhaps it's not that LLMs are inherently bad at IE, but that we haven't found the best way to use them for IE yet. LLMs are also becoming better over time. A blanket statement, like "LLMs are not good few-shot information extractors in general", may need more qualification, like "the **current** LLMs **under vanilla prompting settings** are not good few-shot information extractors in general". This doesn't significantly undermine the contribution of this paper, and I think it's still a worthy paper to be presented at the conference. But it's important to send the most accurate message to the community to inform future research.

Kai Zhang, Bernal Jimenez Gutierrez, Yu Su. Aligning Instruction Tasks Unlocks Large Language Models as Zero-Shot Relation Extractors, Findings of ACL 2023.

---

### Decision · Program_Chairs · 2023-10-07

**Decision:**

Accept-Findings

**Comment:**

This is a comprehensive empirical study that leads to the observation that LLMs are generally not as competitive for IE tasks than fine-tuned small LMs. The paper then proposes a hybrid strategy that uses LLMs to rerank for "hard" examples that a small LM finds challenging. Empirical results demonstrate the effectiveness of this strategy.

Strengths:
- Solid and comprehensive empirical study on an important problem
- Clear and professional writing
- An intuitively plausible hybrid strategy with demonstrated effectiveness

Weaknesses:
- As the reviewers pointed out, the paper falls short in investigating and explaining why LLMs are underperforming, especially given the known sensitivity to prompts. The authors have responded to this criticism with new analysis and experiments. The authors are strongly encouraged to add these to the revised version.
- As Zhang et al. (2023), which is already cited in this paper, have shown, LLMs' RE performance can be substantially improved by reformulating RE as a QA problem. One possibility demonstrated through that work is, perhaps it's not that LLMs are inherently bad at IE, but that we haven't found the best way to use them for IE yet. LLMs are also becoming better over time. A blanket statement, like "LLMs are not good few-shot information extractors in general", may need more qualification, like "the **current** LLMs **under vanilla prompting settings** are not good few-shot information extractors in general". This doesn't significantly undermine the contribution of this paper, and I think it's still a worthy paper to be presented at the conference. But it's important to send the most accurate message to the community to inform future research.

Kai Zhang, Bernal Jimenez Gutierrez, Yu Su. Aligning Instruction Tasks Unlocks Large Language Models as Zero-Shot Relation Extractors, Findings of ACL 2023.